# Single cell RNA sequencing of the adult *Drosophila* eye reveals distinct clusters and novel marker genes for all major cell types

Kelvin Yeung [1,6], Komal Kumar Bollepogu Raja[1,6], Yoon-Kyung Shim[1], Yumei Li [2,3,4], Rui Chen [2,3,4] & Graeme Mardon [1,2,5✉]

The adult *Drosophila* eye is a powerful model system for phototransduction and neurodegeneration research. However, single cell resolution transcriptomic data are lacking for this tissue. We present single cell RNA-seq data on 1-day male and female, 3-day and 7-day old male adult eyes, covering early to mature adult eyes. All major cell types, including photoreceptors, cone and pigment cells in the adult eye were captured and identified. Our data sets identified novel cell type specific marker genes, some of which were validated in vivo. R7 and R8 photoreceptors form clusters that reflect their specific *Rhodopsin* expression and the specific *Rhodopsin* expression by each R7 and R8 cluster is the major determinant to their clustering. The transcriptomic data presented in this report will facilitate a deeper mechanistic understanding of the adult fly eye as a model system.

[1] Department of Pathology and Immunology, Baylor College of Medicine, One Baylor Plaza, Houston, TX 77030, USA. [2] Department of Molecular and Human Genetics, Baylor College of Medicine, One Baylor Plaza, Houston, TX 77030, USA. [3] Human Genome Sequencing Center, Baylor College of Medicine, One Baylor Plaza, Houston, TX 77030, USA. [4] Structural and Computation Biology and Molecular Biophysics, Baylor College of Medicine, One Baylor Plaza, Houston, TX 77030, USA. [5] Program in Developmental Biology, Baylor College of Medicine, One Baylor Plaza, Houston, TX 77030, USA. [6] These authors contributed equally: Kelvin Yeung, Komal Kumar Bollepogu Raja. ✉email: gmardon@bcm.edu

The fruit fly *Drosophila melanogaster* is a powerful model organism that has been instrumental in elucidating conserved mechanisms of cell fate determination, differentiation, proliferation, survival, and growth. Many tools for genetic manipulation were developed for *Drosophila*, including the Gal4/UAS system for expressing any transgene in a time and/or tissue-specific manner and the Flp/FRT system for inducing clones of homozygous mutant tissue in a heterozygous animal. Moreover, *Drosophila* and humans are remarkably conserved and thus biological insights from research in *Drosophila* are often directly relevant to vertebrate development and disease.

The eye is one of the most highly studied tissues in *Drosophila*. Many conserved genetic networks and signaling pathways are required during *Drosophila* eye development and the eye has served as an important system for characterizing these pathways. The adult eye is organized in a highly stereotypical pattern, and any perturbation caused by genetic alteration can be easily scored in living animals, rendering it one of the most powerful tools for genetic screens in higher eukaryotes. In addition, since the eye is not required for survival in a laboratory setting, it is well-suited for testing the function of genes without causing lethality. For these reasons, along with the extensive set of tools for genetic manipulation, the adult *Drosophila* eye has served as an important human disease model, particularly for neurodegenerative diseases. For example, the adult *Drosophila* eye has been used to study neurodegenerative conditions such as Alzheimer, Parkinson, Huntington, and Glutamine repeat disease[1–3].

The adult *Drosophila* eye comprises about 750 repeating units, called ommatidia (Fig. 1a, b). Each ommatidium contains eight photoreceptor cells (named R1-R8), four lens-secreting cone cells, one tertiary, two primary, and three secondary pigment cells, and two bristle cells (Fig. 1c–e). Each photoreceptor detects light with its rhabdomere, a specialized structure made of tens of thousands of microvilli where the light-sensing Rhodopsin proteins are localized. The photoreceptors in each ommatidia are arranged in a stereotypic manner where the rhabdomeres form a trapezoidal shape. Rhabdomeres from R1-6 form the outline of the trapezoid, while the R7 and R8 rhabdomeres occupy the center of each trapezoid, with the R7 rhabdomere positioned directly apical to the R8 rhabdomere. The four cone cells are positioned apical to all the photoreceptors. During pupal development, the cone cells are required to form the lens of the eye[4]. In the adult eye, cone cells function as support cells for photoreceptors[5]. The cone cells are surrounded by the two primary pigment cells. The secondary and tertiary pigment cells are localized to the periphery of each ommatidium and act as optical insulators against scattering light between ommatidia in the adult eye. Pigment cells are also required for the production and transport of chromophore, an essential component of phototransduction in *Drosophila*, into photoreceptors.

Phototransduction has been extensively studied in the adult *Drosophila* eye. Each photoreceptor cell expresses one type of *Rhodopsin* (R1-6 expresses *ninaE*, R7 expresses either *Rh3* or *Rh4*, and R8 expresses *Rh5* or *Rh6*)[6–11]. The outer photoreceptors (R1-6) are specialized to detect motion while the inner photoreceptors (R7 and R8) are specialized for color vision[12,13]. Briefly, in *Drosophila* phototransduction, light first activates Rhodopsins[14]. This causes Rhodopsin to isomerize to Metarhodopsin and activates a Rhodopsin-bound G protein. This in turn activates Phospholipase C and triggers the downstream activation

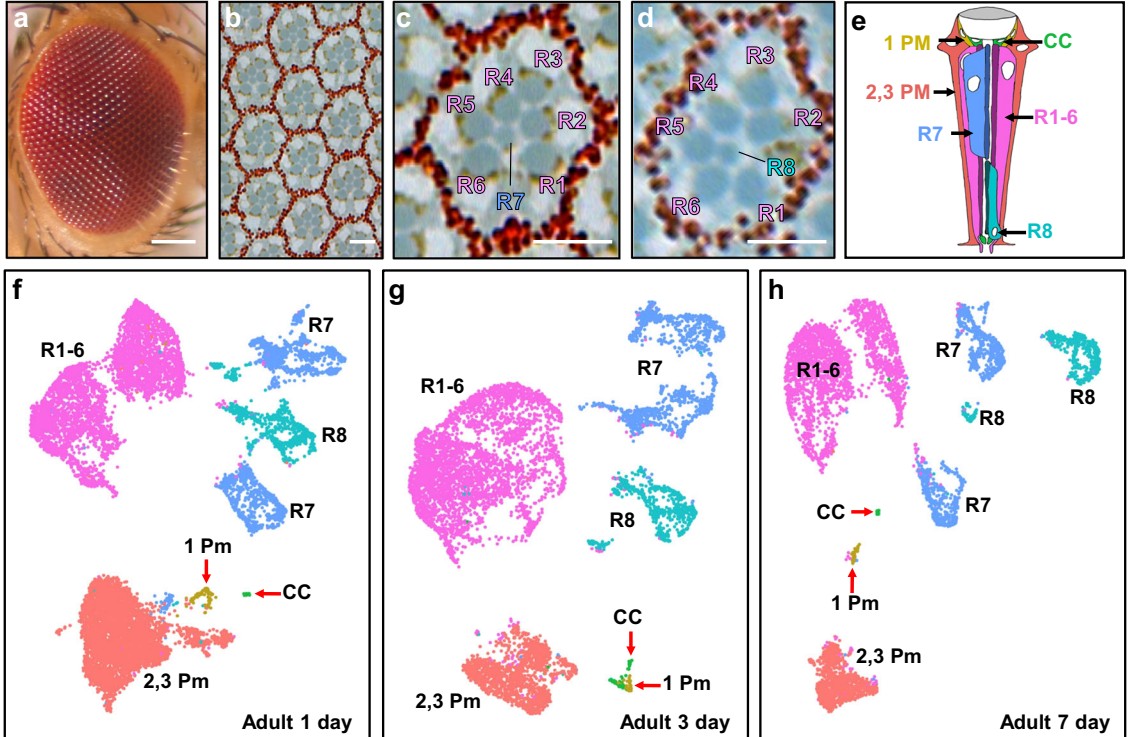

**Fig. 1 Single-cell RNA sequencing of adult *Drosophila* eyes reveals all expected cell types. a** External image of an adult eye; scale bar: 100 μm. **b** Tangential section of an adult eye shows about 10 ommatidia; scale bar: 5 μm. **c, d** High magnification views of tangential sections of an ommatidium at the R7 (**c**) and R8 (**d**) focal planes. Scale bars: 5 μm. **e** Schematic of the different cell types present in an adult ommatidium; there are six R1-6 cells in each ommatidium but only two are shown in this view. Reproduced/adapted with permission from Fig. 1b in Rister et al., 2013, *Development*[86] (https://doi.org/10.1242/dev.079095). Teal: R8; purple: R1-6; blue: R7; orange: secondary and tertiary pigment cells; yellow: primary pigment cells; green: cone cells. **f–h** Seurat UMAP cluster plots of scRNA-seq results of 1-day (**f**), 3-day (**g**), and 7-day (**h**) old adult eyes. Cell cluster identities: R1-6, R7, and R8 are photoreceptors R1-8; 1 Pm are primary pigment cells; 2,3 Pm are secondary and tertiary pigment cells; and CC are cone cells.

**Table 1 Quality control metrics for adult eye scRNA-seq data sets.**

| Metrics | 1-Day old ♂ | 3-Day old ♂ | 7-Day old ♂ | 1-Day old ♀ |
|---|---|---|---|---|
| Total reads | 583 M | 267 M | 104 M | 571 M |
| Total genes | 10,874 | 10,214 | 9619 | 10.297 |
| Estimated number of cells | 16,275 | 8701 | 6335 | 8765 |
| Median genes per cell | 938 | 855 | 618 | 1008 |
| Median UMI per cell | 3161 | 3168 | 1872 | 4465 |
| Fraction of reads in cells | 87.2% | 87.7% | 80.8% | 87.6% |

**Table 2 Number of cells by cell type in each data set.**

| Cell type | 1-Day ♂ | 3-Day ♂ | 7-Day ♂ | 1-Day ♀ | [a]Expected % |
|---|---|---|---|---|---|
| R1-6 | 4079 (37%) | 3538 (50%) | 2460 (49%) | 3156 (55%) | 30% |
| R7 | 1770 (16%) | 1251 (18%) | 953 (19%) | 965 (17%) | 5% |
| R8 | 791 (7.1%) | 575 (8.2%) | 504 (10%) | 386 (7.4%) | 5% |
| Cone cells | 28 (0.25%) | 56 (0.80%) | 24 (0.48%) | 0 (0%) | 20% |
| 1° Pigment | 122 (1.1%) | 44 (0.63%) | 48 (0.95%) | 0 (0%) | 10% |
| 2°, 3° Pigment | 4374 (39%) | 1551 (22%) | 1046 (21%) | 1219 (21%) | 20% |
| Total | 11,164 | 7015 | 5035 | 5726 | 90% |

[a]Expected % does not add up to 100% as bristle cells are not counted.

of two cation channels, causing an influx of cations into the photoreceptors. Despite some differences between the *Drosophila* and vertebrate phototransduction pathways, the general mechanism involves Rhodopsin being activated by light, which triggers a G protein signaling cascade leading to the activation of cation channels and an influx of cations into the photoreceptor. Thus, insights into phototransduction from *Drosophila* eye studies are also applicable to vertebrate phototransduction.

Although the *Drosophila* eye is well characterized and commonly used as a disease model, we are still far from a complete understanding of the eye. Whole tissue transcriptome data of adult eyes were previously reported but the resulting data mixes the transcriptomes of neuronal photoreceptor cells and non-neuronal pigment cells and cone cells. The mixing of these very different cell types confounds the interpretation of the transcriptome data. Although single-cell RNA-sequencing has been employed in *Drosophila* at multiple developmental stages and tissues to obtain single-cell resolution transcriptomic data, high-resolution data are still lacking for the adult eye[15–17]. Single-cell transcriptome data from the adult eye provides a detailed view of the transcription landscape of each cell type. This in turn can afford a far more detailed dataset for unraveling the molecular mechanisms of eye development and disease models. Single-cell transcription data can also lead to the rapid discovery of new genes with cell-type-specific functions.

In this report, we present single-cell transcriptome data of more than 27,000 cells prepared from intact adult *Drosophila* eye cells from 1-day, 3-day, and 7-day-old male and 1-day-old female animals. These three time points cover the transcriptomes of the early and mature adult eye. All major cell types of the adult eye cluster distinctly in all three time points and the identities of all major cell types and novel cell-type markers discovered from these data sets were validated in vivo. Our data sets show a clear distinction between R1-6, R7, and R8 photoreceptor subtypes and that the expression of cell type-specific *Rhodopsins* is the main contributor to these distinctions. These data also reveal many novel markers for each major cell type. We expect these data to greatly aid in gene discovery, reagent development, and a high-resolution mechanistic understanding of the adult fly eye as a model system.

## Results

**Single-cell RNA-sequencing reveals the transcriptomes of photoreceptors, cone, and pigment cells of 1-day, 3-day, and 7-day-old adult eyes.** We performed single-cell RNA-sequencing (scRNA-seq) on dissociated 1-day (1D), 3-day (3D), and 7-day (7D)-old male *Canton-S* adult eyes using 10x Genomics Chromium, a droplet-based single-cell sequencing platform. Males were chosen specifically so that genes expressed from the Y chromosome would also be captured. To avoid any stress-related transcriptional responses, we dissected and dissociated all samples in the presence of a transcription-inhibiting drug, Actinomycin D[18]. Overall, we obtained a large number of cells compared to the complexity of the tissue and each preparation had greater than 80% viability. Specifically, prior to cell filtering, we had more than 6300 cells from each time point, with more than 100 million reads and more than 600 median genes detected per cell (Table 1). The fraction of reads in cells, a metric that reflects cell viability, is more than 80% for all three time points.

Low-quality cells with high mitochondrial gene expression, droplets with potential multiple cells, non-eye lamina neurons that express neural-specific markers, and glial cells that express glial-specific markers were all filtered out to yield an eye-specific data set. Ambient RNA was removed using SoupX[19]. After filtering, we have more than 5000 cells for each time point (Table 2). The three filtered datasets were individually analyzed with Seurat v4[20–22]. Using known cell type-specific markers, each major cell type was identified and annotated on Seurat UMAP cluster plots (Fig. 1f–h). Cell clusters for each photoreceptor subtype, R1 to R8, and cell clusters for primary, secondary, and tertiary pigment cells and cone cells are present in the filtered datasets from all three time points. For each time point, we have captured over 3900 R1-8 photoreceptor cells and over 1000 pigment cells (Table 2). Although cone cells and primary pigment cells are underrepresented in our filtered or unfiltered data sets from all three time points, all other major cell types (R1-8 photoreceptors, secondary and tertiary pigment cells) are well represented.

We performed pseudotime analyses with Monocle 3 on the three datasets to check for any transcriptomic changes as the eye ages[23,24]. However, Monocle 3 did not identify any trends

between the 1D, 3D, and 7D datasets (Supplementary Fig. 1). Specifically, each cell type cluster and partition identified by Monocle 3 contains cells from all three time points. There are no cell clusters that contain cells only from a single time point. In addition, there is no clear segregation of cells from each time point within each cell type cluster. These results suggest that the transcriptomes of adult eyes do not change extensively as the eye ages from 1 to 7 days. However, we notice a decrease in the number of genes and number transcripts as the animal ages (Supplementary Fig. 1c, d). This is consistent with studies of the aging *Drosophila* brain where it was reported that as the animal ages, the number of genes expressed and number of transcripts decrease[25].

**Identification of R8 cell clusters and *CG2082* as an R8 marker gene.** We identified R8 cell clusters using expression of the known R8-specific markers *Rhodopsin 5* (*Rh5*), *Rhodopsin 6* (*Rh6*), and *senseless* (*sens*)[6,7,26] (7D data are shown in Fig. 2a–d; 1D and 3D are in Supplementary Fig. 2a–d). Using these markers, the annotated R8 cell clusters appear as 2 distinct groups in all time points. Although *Rh5* and *Rh6* are R8-specific *Rhodopsins*, they are not expressed in all R8 photoreceptors. Specifically, each R8 outside the dorsal periphery of the eye, expresses either *Rh5* or *Rh6* but not both *Rhodopsins*. *Rh5*-expressing R8 cells are named 'pale' R8s while *Rh6* expressing R8s are termed 'yellow'[27,28]. In the adult eye, the expected ratio of pale R8s to yellow R8s is about 30:70. We confirmed this is the case with our *Canton-S* stock by staining adult eyes with Rh5 and Rh6 antibodies. Our *Canton-S* pale R8:yellow R8 ratio is about 30:70 (419 pale:1156 yellow R8s, Supplementary Fig. 2g shows an example *Canton-S* R8 image). However, we observed that the ratios of scRNA-seq captured pale to yellow R8s range from 10:90 to 14:86 in our data sets (Supplementary Fig. 2f). The number of pale R8 cells appears to be underrepresented. Only 58–113 pale R8s were captured as opposed to 441–678 yellow R8s captured. Despite these cell number discrepancies, pale and yellow R8s clusters are well separated and *Rh5* and *Rh6* are strongly expressed in their corresponding clusters while *sens* is weakly expressed in both clusters (Fig. 2b–d and Supplementary Fig. 2b–d). The separation of *Rh5* and *Rh6* expressing cells within the R8 cell cluster is consistent with the exclusivity of Rh5 and Rh6 expression in adult R8s in vivo[29]. In summary, the expression of all three R8 marker genes supports the R8 annotation of these cell clusters for all three time points.

To identify marker genes, we first performed differential gene expression analyses with Seurat. Seurat-called marker gene lists were then screened with FeaturePlots to assess the specificity of each gene in its corresponding cell type. Since cone cell and primary pigment cell clusters have very few cells, these clusters were ignored in the FeaturePlot screening of R1-6, R7, R8, and secondary and tertiary pigment cell genes. For screening called cone cell marker genes, we grouped primary pigment cells with secondary and tertiary pigment cells. We treated cone cell clusters as a separate cluster while screening the called primary pigment cell marker genes. We classified specificity into three different classes: (1) contains genes that are only expressed in the corresponding cell type; (2) contains genes that are expressed in the corresponding cell type and one other cell type; (3) contains genes that expressed in the corresponding cell type and more than one other cell type but these genes are not expressed ubiquitously.

Differential gene expression analyses of R8 cell clusters from all three time points reveal 104 to 261 Seurat-called R8 marker genes (Supplementary Data 1). Of these called R8 markers, 83 genes are common to all three time points. FeaturePlot analyses were performed on all called R8 marker genes to determine their R8

expression and specificity. We found 77 genes were expressed in R8 (Class 1, 2, or 3 in specificity) and 8 are Class 1 specific. An example of an R8-specific marker gene is *CG2082*. FeaturePlots of *CG2082* show that it is expressed in R8 cell clusters in all three time points (Fig. 2e and Supplementary Fig. 2e). To test if *CG2082* is expressed in this pattern in the adult eye in vivo, we use the *Trojan-Gal4* (*T2A-Gal4*) system which drives Gal4 expression in a pattern that usually accurately reflects the transcription of the gene in which the *T2A-Gal4* cassette is inserted[30,31]. To visualize *CG2082* expression in R8 cells, we used a membrane bound GFP reporter (*UAS-mCD8-GFP*) driven by *CG2082-T2A-Gal4*. These eyes were costained with anti-Rh5 and anti-Rh6 antibodies. The results show that GFP is present in membranes of cells that also express Rh5 or Rh6 (Fig. 2f, white arrow and arrowhead point to colocalization of GFP and Rh5 or Rh6, respectively). In addition to using a membrane localizing reporter, a nuclear localizing reporter (*UAS-mCherry-nls*) driven by *CG2082-T2A-Gal4* also shows specific reporter expression in R8 nuclei (Fig. 2g). *CG2082-T2A Gal4* driven mCherry only shows mCherry expression in R8 cells. mCherry is not present in the R1-7 photoreceptors (Fig. 2g, white bracket) nor the pigment cells. To our knowledge, *CG2082* is previously uncharacterized and its expression in adult eyes has not been reported in vivo. These results suggest that *CG2082* is specifically expressed in R8 cells and is a R8 marker gene in the adult eye.

**Identification of R7 cell clusters and *igl* is an R7/8 marker gene.** R7 photoreceptor cell clusters express R7-specific Rhodopsins, *Rhodopsin 3* (*Rh3*) and *Rhodopsin 4* (*Rh4*), and a known adult R7 marker *prospero* (*pros*) in all three time points (Fig. 2i–l and Supplementary Fig. 3a–d)[8–10,32]. Similar to the exclusive pattern of expression of *Rh5* and *Rh6* in R8s, most R7s also express either *Rh3* or *Rh4* but not both. In the eye, the *Rh3*-expressing R7s (pale R7s) are paired with *Rh5*-expressing R8s (pale R8) while *Rh4*-expressing R7s (yellow R7s) are paired with *Rh6*-expressing R8s (yellow R8) within the same ommatidium[27,28]. An exception is in the dorsal third of the eye, where yellow R7s in that region express both *Rh3* and *Rh4*[33]. Due to the 1:1 pairing of pale R7/8 and yellow R7/8, the ratio of pale:yellow R7s is about 30:70. Immunofluorescence staining with Rh3 and Rh6 of our *Canton-S* adult eyes show the expected ratio of pale to yellow ommatidia (ratio of pale to yellow is about 30:70, 244 Rh3:440 Rh6). However, in our scRNA-seq data sets, the observed pale:yellow R7 ratios are about 50:50 (Supplementary Fig. 3f). The number of captured pale and yellow R7 cells do not appear to be underrepresented in our data sets. Despite the ratio discrepancies, our scRNA-seq data from all three time points show that R7 cells can be separated into two major groups: one predominantly comprises *Rh3*-expressing pale R7 cells, while the other contains *Rh4*-expressing yellow R7 cells (which also include the dorsal third R7s) (Fig. 2j, k and Supplementary Fig. 3c, d, red arrows point to dorsal third yellow R7s). In contrast to *Rh3* and *Rh4*, *pros* is expressed in all R7 cells (Fig. 2l). The clustering reflects the *Rh3* and *Rh4* expressing R7 cells in vivo. The expression of *Rh3*, *Rh4*, and *pros* support the R7 cluster annotation in all three time points.

Differential gene expression analyses of R7 cell clusters from all time points reveal 90 to 311 R7 marker genes and 78 R7 marker genes are found in all three time points (Supplementary Data 2). FeaturePlot analyses of all called R7 marker genes revealed 86 genes in Class 1, 2, or 3. Only 5 called markers (*Rh3*, *Rh4* and *pros* included) are Class 1 specific. However, during our FeaturePlot analyses of R7 and R8 marker genes, we observed that 24 genes are expressed only in R7 and R8 cells. Thus, they are potential R7/8 marker genes. An example of a novel R7/8 marker is *igloo*

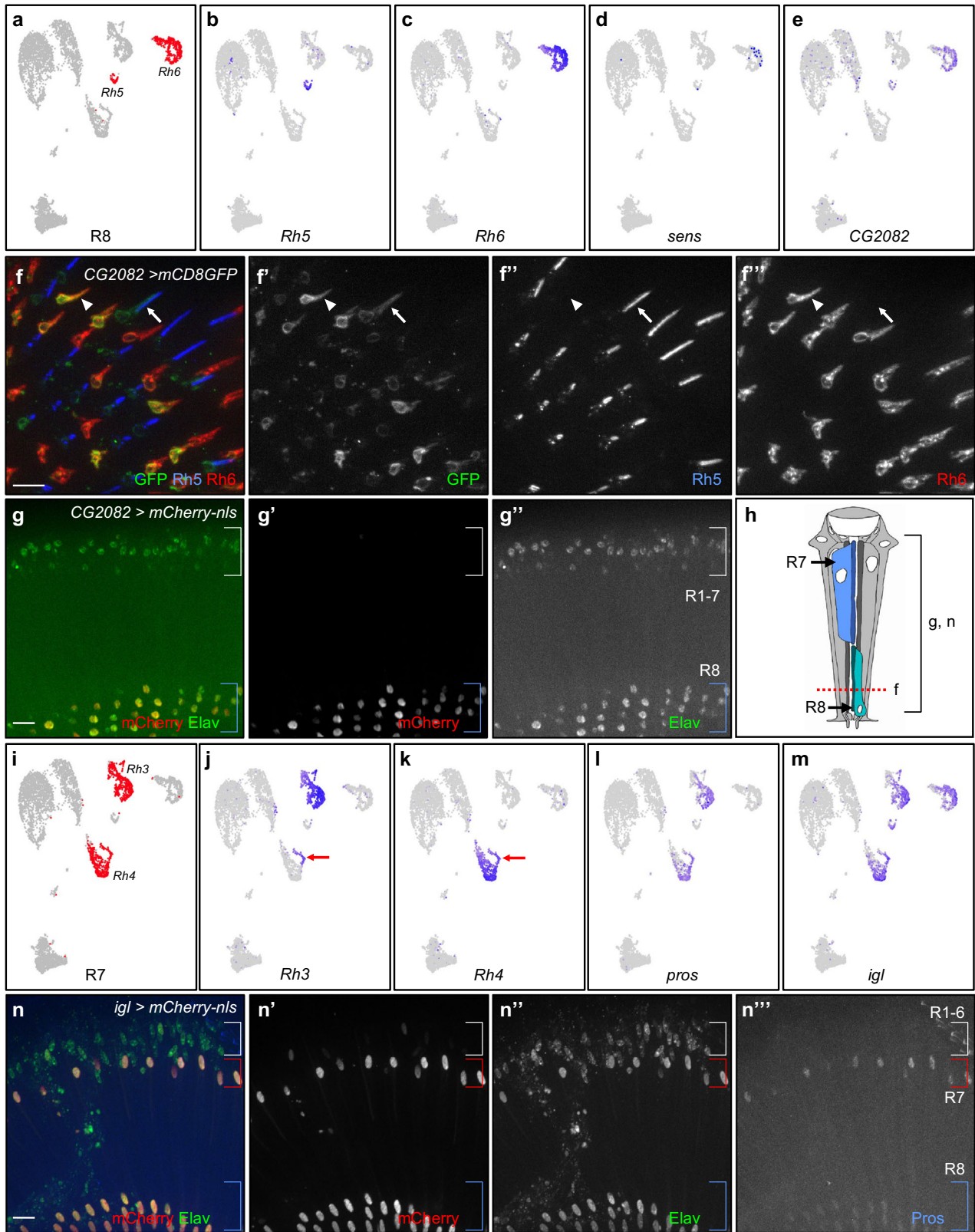

(*igl*, Fig. 2m, Supplementary Fig. 3e). To our knowledge, *igl* expression in adult R7 and R8 was not previously reported in vivo. To test if *igl* is specifically expressed in R7 and R8 in vivo, mCherry-nls was driven with *igl-T2A-Gal4*. Immunofluorescence staining shows that *igl* driven mCherry is specifically expressed in both R7 and R8 in the eye (Fig. 2n). To our knowledge, *igl* has not been shown to be expressed in adult R7 and R8 photoreceptors.

These results suggest that *igl* is a novel marker for R7 and R8 photoreceptors in the adult eye.

**Presence of specialized R7 and R8 cell types in adult eye scRNA-seq.** In the dorsal third of the adult eye, yellow R7 cells express both *Rh3* and *Rh4*[33]. FeaturePlots of our scRNA-seq data

**Fig. 2 Identification of R8 and R7 photoreceptor clusters and novel markers. a** Seurat UMAP cluster plots of 7-day-old adult eyes with R8 cell clusters highlighted in red. **b–e** FeaturePlots showing gene expression on UMAP cluster plots for *Rh5* (**b**), *Rh6* (**c**), *sens* (**d**) and *CG2082* (**e**). All cells that express *sens* and *CG2082* were brought to the front in FeaturePlots (**d**) and (**e**). **f** Tangential view immunofluorescence images of a 7-day-old adult *CG2082-T2A-Gal4 > UAS-mCD8-GFP* eye costained with GFP (green), Rh5 (blue), and Rh6 (red). White arrows mark an R8 rhabdomere where GFP colocalizes with Rh5 and white arrowheads mark an R8 rhabdomere where GFP colocalizes with Rh6. **g** Coronal view immunofluorescence images of a 7-day-old adult *CG2082-T2A-Gal4 > UAS-mCherry-nls* eye costained with mCherry (red) and Elav (green). Blue brackets mark R8 nuclei and white brackets mark R1-7 nuclei. **h** Schematic drawing of the different cell types of an ommatidium, where the red dashed line marks the focal plane of the tangential view of (**f**) and a bracket marks the views shown in (**g**) and (**n**). **i** Seurat UMAP cluster plots of 7-day-old adult eyes with R7 cell clusters highlighted in red. **j–m** FeaturePlots showing gene expression for *Rh3* (**j**), *Rh4* (**k**), *pros* (**l**), and *igl* (**m**). All cells that express *pros* and *igl* were brought to the front in FeaturePlots (**l**) and (**m**). Red arrows point to the dorsal third yellow R7s which express both Rh3 and Rh4. **n** Coronal view immunofluorescence images of 7-day-old adult *igl-T2A-Gal4 > UAS-mCherry-nls* eye costained with GFP, Elav, Pros. All scale bars: 10 μm.

contain small groups of cells that express both *Rh3* and *Rh4* in all three time points (Fig. 2j, k, Supplementary Fig. 3c, d, red arrows). These cells also express *pros*, a R7 marker (Fig. 2l, Supplementary Fig. 3b). Although differential gene expression and FeaturePlot analyses did not identify any genes that are exclusively expressed in dorsal third yellow R7 cells, Seurat called *Rh3*, *Rh4* and *pros* as the top markers (ranked by *p*-value). The presence of all three genes in the marker list and FeaturePlots suggests that our scRNA-seq data captured dorsal third yellow R7 cells.

Another group of specialized R7s and R8s is the dorsal rim area (DRA) of the eye. This is the row of ommatidia that reside at the dorsal edge of eye. Both R7 and R8 cells in the DRA express *Rh3* and do not express any of the other *Rhodopsins*[26]. DRA R7s and R8s also express *homothorax* (*hth*). Our scRNA-seq data reveal a small group of *Rh3*-positive cells that also express *hth* (Supplementary Fig. 4). The presence of both marker genes suggests that these cells are DRA R7s and R8s. In addition, differential expression and FeaturePlot analyses show that *Skeletor* is a potential new DRA R7 and R8 marker (Supplementary Fig. 4).

**Identification of R1-6 cell clusters and *Zasp66* as a novel photoreceptor marker gene.** R1-6 cell clusters from all three time points are identified by their strong expression of *ninaE*, the *Rhodopsin* specifically expressed by R1-6, in combination with the low or absent expression of *Rh3-6* (Figs. 2b, c, j, k, 3a, b, f and Supplementary Fig. 5)[11,34]. The high *ninaE* expression in R1-6 cluster is most obvious in ViolinPlots (Fig. 3f, Supplementary Fig. 5e, f). These cell clusters also express *ocelliless* (*oc*), which is expressed in all adult photoreceptors (Fig. 3c)[35]. Differential gene expression analyses of R1-6 cells found 144-493 Seurat-called marker genes and 130 are common to all three time points (Supplementary Data 3).

FeaturePlot analyses of all R1-6 marker genes show that none of the R1-6 marker genes are exclusively expressed in R1-6. However, we found 76 genes that are expressed in all photoreceptors but not in pigment cells or cone cells clusters in all three time points (Class 3). This observation suggests that the R1-6 marker genes called by Seurat FindAllMarkers function are photoreceptor-specific and may not be specific to R1-6. An example of a photoreceptor-specific marker gene is *Zasp66* (Fig. 3d and Supplementary Fig. 5d). Note that FeaturePlots show *Zasp66* is expressed the strongest in all photoreceptor cells in 1-day and its expression is less strong in the other time points (Supplementary Fig. 5d). To test if *Zasp66* is expressed specifically in photoreceptors in vivo, *Zasp66-T2A-Gal4* was used to drive the mCherry-nls reporter in adult eyes. *Zasp66-T2A-Gal4* driven mCherry is present in all photoreceptor nuclei (Fig. 3g, h). Specifically, tangential view shows mCherry is present in all six R1-6 photoreceptors and coronal view shows mCherry is also present in R7 and R8 photoreceptors. The mCherry staining

colocalizes with Elav, a known nuclear photoreceptor marker, but not with Cut, a cone cell marker[36]. To our knowledge, *Zasp66* has not been shown to be expressed in photoreceptor cells in vivo. Therefore, our results show that *Zasp66* is expressed in all photoreceptor cells and is a novel photoreceptor marker.

**Identification of cone cell clusters and *CG5597* is a novel cone cell marker gene.** Although there were very few cone cells captured (ranging from 24 to 56, much lower than expected, Table 2), they still form a cone cell-specific cluster in all three time points (Fig. 4a, Supplementary Fig. 6a). These clusters strongly express the cone cell marker *cut* (*ct*, Fig. 4b, Supplementary Fig. 6b)[36]. In addition, they express *Crystallin* (*Crys*), which is a major component of the *Drosophila* lens and is expressed by pupal cone cells (Fig. 4c, Supplementary Fig. 6c)[4,37]. Differential expression analyses identified 210 to 753 Seurat-called cone cell marker genes, where 125 are common to all three time points (Supplementary Data 4). FeaturePlot analyses of all Seurat-called cone cell marker genes identified 233 genes which are expressed in cone cells (Class 1, 2, or 3) and 41 of them are Class 1 specific. An example of a novel cone cell-specific marker gene is *CG5597*, an uncharacterized gene (Fig. 4d, Supplementary Fig. 6d). To test if *CG5597* is expressed specifically in adult cone cells, we drove mCherry-nls reporter with *CG5597-T2A-Gal4* in adult eyes. Immunostaining results show that *CG5597* driven *mCherry* is present in cone cell nuclei, where they colocalize with Ct (Fig. 4f). *CG5597* driven *mCherry* is also not expressed in other cell types in the adult eye. Tangential images of *CG5597* driven *mCherry* show mCherry is present in all four cone cells per ommatidium (Fig. 4g). These results support that our identification of the cone cell cluster and that *CG5597* is a novel cone cell marker in the adult eye.

**Identification of pigment cell clusters and novel pigment cell markers.** Proteins encoded by *white* (*w*) and *Photoreceptor dehydrogenase* (*Pdh*) were previously shown to be present in adult eye pigment cells[38,39] and we used these two marker genes to identify pigment cell clusters in our datasets (Fig. 5a–c, f, Supplementary Fig. 7a–e). Our FeaturePlot analyses also show that other genes involved in pigment production, such as *sepia* and *Punch*, are expressed in the pigment cell clusters (Supplementary Fig. 8)[40,41]. *Pdh*-positive cells are present in two separate clusters in each time point. We determined that the larger cluster comprises secondary and tertiary pigment cells, while the smaller cluster represents primary pigment cells (see below). Differential gene expression analyses on the secondary and tertiary pigment cell clusters reveals 332–394 Seurat-called marker genes among which 282 are common to all three time points (Supplementary Data 5). FeaturePlot analyses of all secondary and tertiary pigment cell markers identified 198 genes with Class 1, 2, or 3 specificity. Seurat called 106–369 primary pigment cell marker genes and 79 are common to all three time points (Supplementary

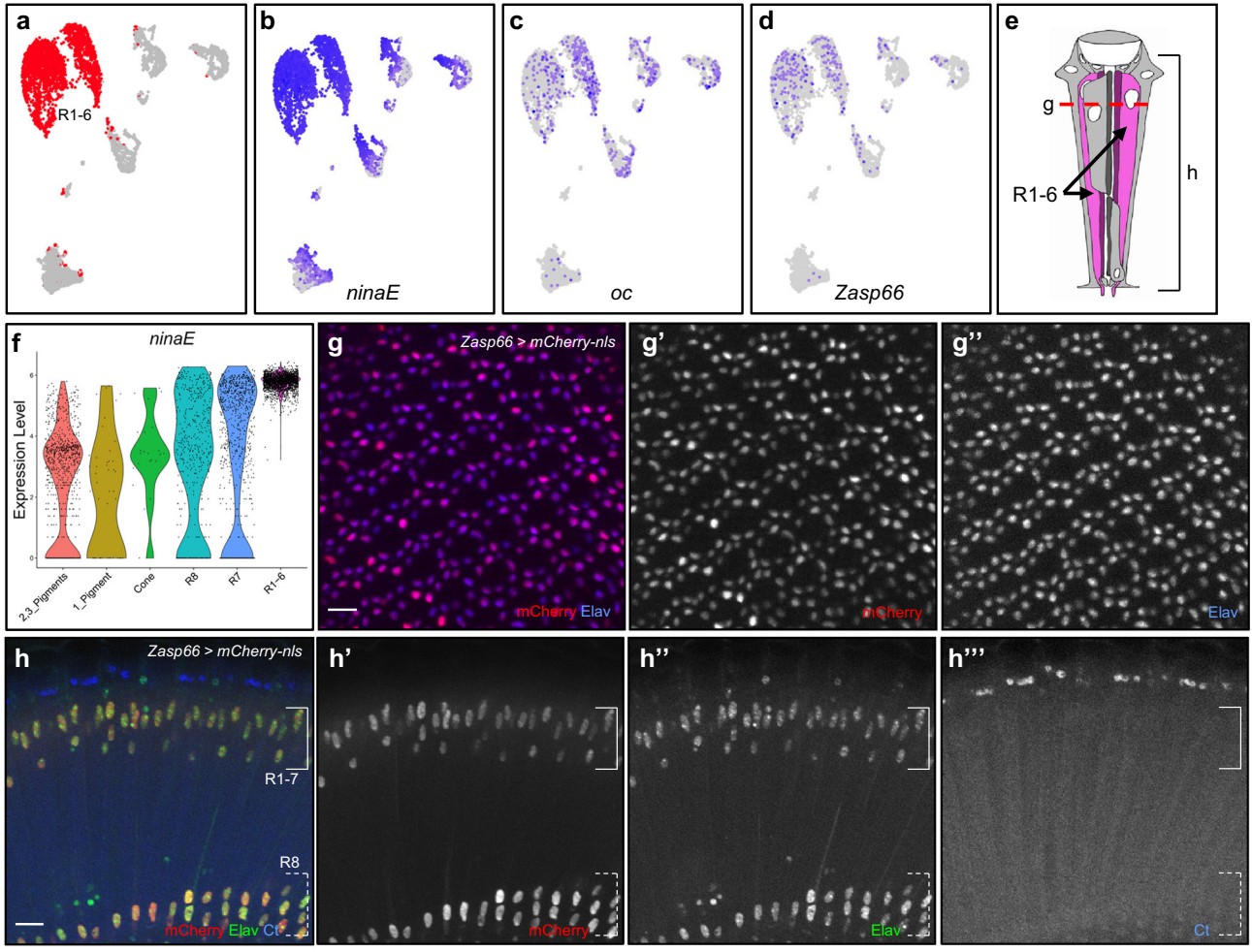

**Fig. 3 R1-6 photoreceptor cluster identification and *Zasp66* is a novel marker for adult photoreceptor cells. a** R1-6 cell clusters in Seurat UMAP cluster plots of 7-day-old adult eyes are colored red. **b–d** FeaturePlots showing gene expression of *ninaE* (**b**), *oc* (**c**), and *Zasp66* (**d**). All cells expressing *oc* and *Zasp66* were brought to the front in FeaturePlots (**c**), (**d**). **e** Schematic of an ommatidium where the bracket and red dotted line mark the focal plane of the views in (**g**) and (**h**), respectively. **f** ViolinPlot of log-normalized *ninaE* expression in all cell types. **g** Tangential view of a 7-day-old *Zasp66-T2A-Gal4 > UAS-mCherry-nls* adult eye costained with mCherry and Elav in red and blue, respectively. **h** Coronal view of 7-day-old *Zasp66-T2A-Gal4 > UAS-mCherry-nls* adult eye costained with Elav and Ct. Solid white bracket marks the R1-7 nuclei and dashed white bracket marks R8 nuclei. Scale bars: 10 μm.

Data 6). FeaturePlot analyses of all primary pigment cell markers showed 227 genes with Class 1, 2, or 3 specificity.

An example of a novel secondary and tertiary pigment cell-specific marker identified from our scRNA-seq data is *santa-maria*. FeaturePlot analyses show that *santa-maria* is expressed in most cells of the secondary and tertiary pigment cell clusters from all three time points with 7D showing the most specific expression (Fig. 5d and Supplementary Fig. 7f). Staining of 7-day-old eyes from *santa-maria-T2A-Gal4* driven mCherry-nls animals shows mCherry expression in cells that do not express Elav (Fig. 5g, h). The mCherry positive nuclei are arranged on the periphery of each ommatidium and these nuclei are apical to the photoreceptor nuclei. The lack of overlap between mCherry and Elav and the peripheral and apical positioning of the mCherry suggest that *santa-maria* is expressed in the secondary and tertiary pigment cells. Thus *santa-maria* is an example of a novel marker of secondary and tertiary pigment cells identified in our data sets.

The pigment cell clusters from all three time points comprise a large cluster of cells and a smaller cluster (Fig. 5a and Supplementary Fig. 7a). While the smaller subclusters weakly express or do not express *w*, they do express *Pdh* in all time points, which suggests that they may be pigment cells. We noted that the number of primary pigment cells captured were much lower than expected (captured 44–122 primary pigment cells, Table 2). FeaturePlot analyses reveal *wrapper* expression in the smaller group, but not in the larger group, at each time point (Fig. 5e and Supplementary Fig. 7g). Adult eyes with *mCherry-nls* driven by *wrapper-T2A-Gal4* show mCherry expression in cells apical to photoreceptor cells (Fig. 5i). The tangential view of *wrapper-T2A-Gal4 > mCherry-nls* shows two mCherry positive nuclei per ommatidium (Fig. 5j). The position and number of mCherry nuclei suggest that these nuclei may be primary pigment cells, of which there are only two per ommatidium. To exclude the possibility that *wrapper-T2A-Gal4 > mCherry-nls* nuclei are cone cell nuclei, *wrapper-T2A-Gal4* driven mCherry eyes were costained with Ct (Fig. 5j, k) and found that *wrapper*-driven mCherry nuclei do not express Ct. We also drove *mCherry-nls* with both *wrapper-* and *santa-maria-T2A-Gal4*. When compared with *santa-maria-T2A-Gal4 > mCherry*, we observed 2 additional apically positioned, mCherry-positive nuclei per ommatidium for the double *T2A-Gal4* eyes (Supplementary Fig. 7h). This further suggests that the pigment cells expressing *wrapper* are primary pigment cells, not secondary or tertiary pigment cells. Conversely, *santa-maria* is expressed in secondary and tertiary pigment cells but not primary pigment cells. Taken together, these data suggest

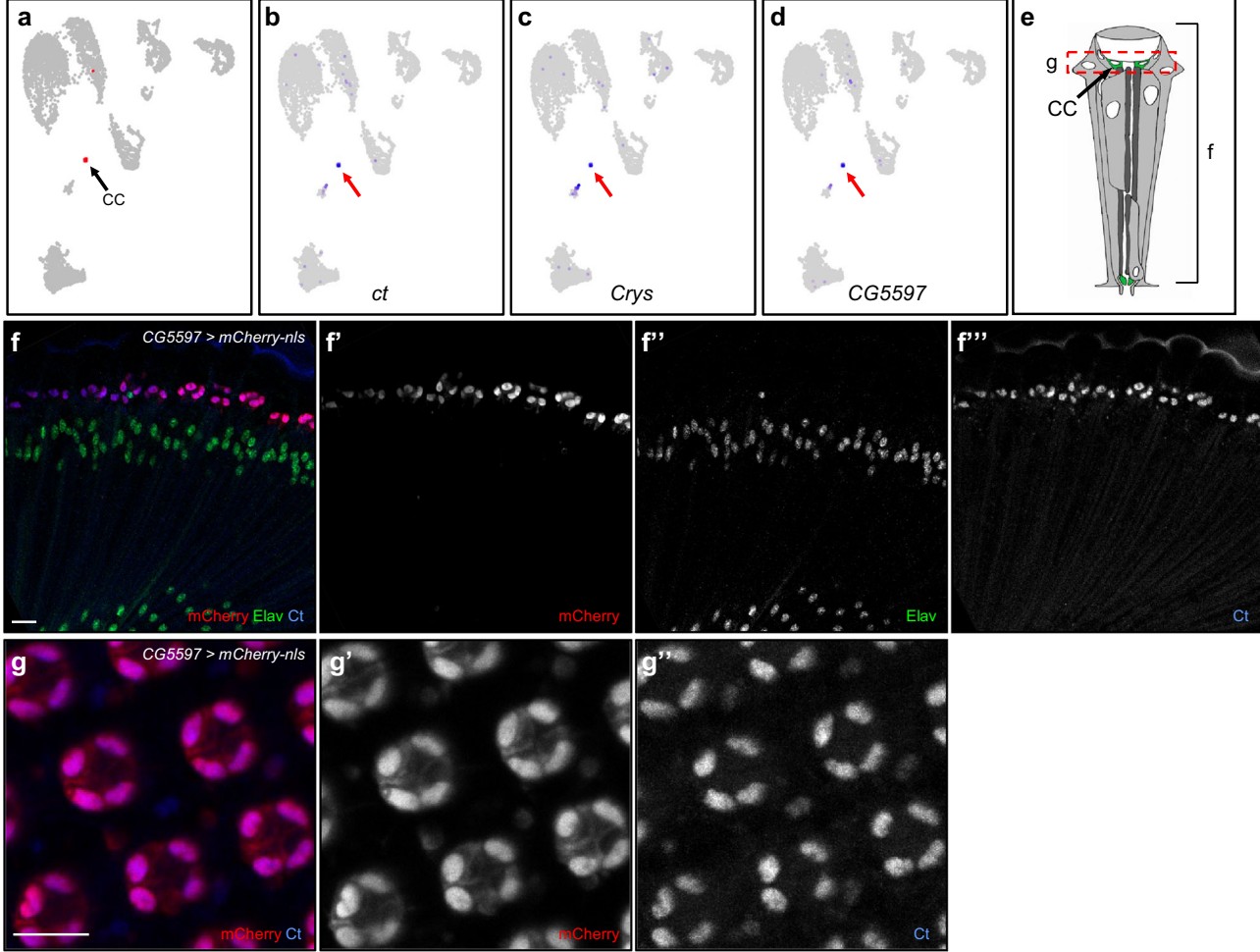

**Fig. 4 Cone cell cluster identification and _CG5597_ is a novel marker for adult cone cells. a** Cone cell cluster (CC, arrow) in Seurat UMAP cluster plots of 7-day-old adult eyes are colored red. **b–d** FeaturePlots showing expression of _ct_ (**b**), _Crys_ (**c**), and _CG5597_ (**d**). All expressing cells are brought to the front. Red arrows point to the cone cell clusters. **e** Schematic of an ommatidium showing the position of the tangential view shown in (**f**) and coronal view shown in (**g**). **f, g** Immunostaining images of a _CG5597-T2A-Gal4 > UAS-mCherry-nls_ 7-day-old adult eye costained with Ct and Elav with coronal view shown in (**f**) and tangential view shown in (**g**). Scale bars: 10 μm.

that _wrapper_ is not expressed in cone cells but it is expressed in primary pigment cells. Moreover, _wrapper_ and _santa-maria_ are novel markers for primary pigment cells and secondary and tertiary pigment cells, respectively.

**Adult male and female photoreceptor and pigment cell transcriptomes are highly similar**. In addition to male adult eyes, we performed scRNA-seq on 1-day-old female adult eyes (Fig. 6a). The quality control metrics of the female adult eye data are similar to the male data sets (Table 1). The female data was filtered and analyzed in the same way as the male data sets and the filtered female data has 5726 cells. The female scRNA-seq identifies secondary and tertiary pigment cells and all photoreceptors (R1-6, R7, and R8) but cone cells or primary pigment cells were not readily identified. Dorsal third R7 cells and DRA R7 and R8 cells can also be identified in the adult female data set (Supplementary Fig. 9). The most abundant female cell type is R1-6, followed by pigment cells, R7 and R8 (Table 2).

To further compare female and male 1D adult eyes, we combined the two data sets using Harmony (Fig. 6b)[42]. This algorithm removes batch effects, integrates the two data sets, and reclusters the total data as one data set. The merged dataset totals 16,890 cells (Table 3). The integrated Seurat UMAP cluster plot

shows the same major cell type clusters (R1-6, R7, R8, cone cells, and primary to tertiary pigment cells) as the male only data sets (Fig. 6b). With the exception of cone cells and primary pigment cells, each cell type cluster is well represented by both female and male cells (Fig. 6b–d, Table 3). All photoreceptors and secondary and tertiary pigment cell clusters are not biased for female or male cells in the integrated dataset. We also compared the number of female and male cells per cluster in the Harmony integrated output with the number of female and male cells per cluster prior to Harmony. The number of female and male cells in each major cell type cluster is also approximately equal to the cell numbers from the female or male only data sets, respectively (Tables 2 and 3). Although the female-only clustering does not readily show primary pigment or cone cell clusters, the Harmony integrated data set assigned 33 female cells to the primary pigment cell cluster and six female cells to the cone cell clusters. It is possible that there were too few female primary pigment and cone cells in the female-only data to be segregated as distinct clusters and thus they were not visible in the female-only data set. Alternatively, these 39 female cells may be misassigned in Harmony. This may be the case as these female cells were originally grouped in the R1-6 cluster in the female-only data set. FeaturePlots show cell type-specific marker genes are expressed in the expected clusters in the Harmony integrated data set,

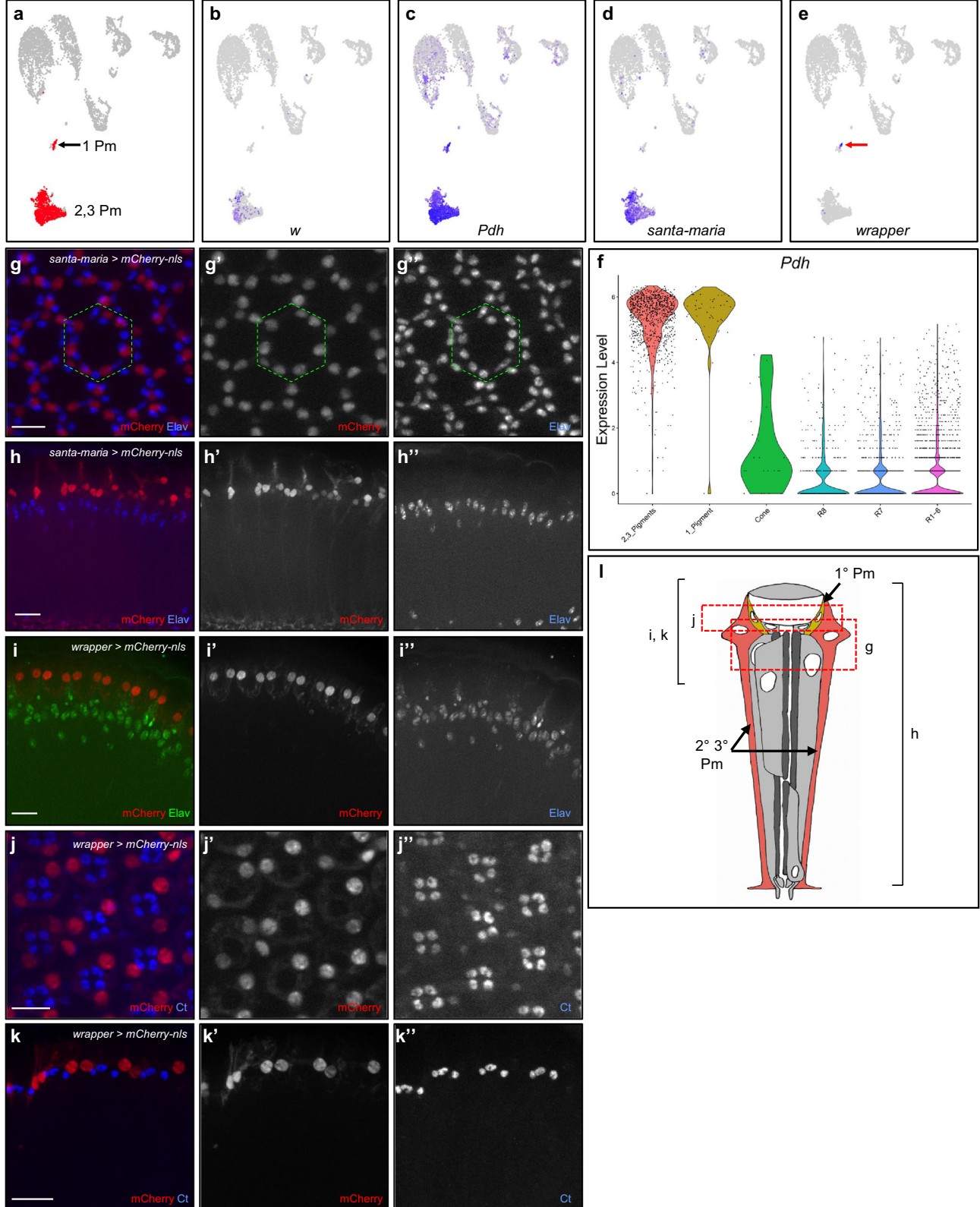

indicating the correct cluster annotation and assignment. Therefore these observations show most male and female cells were assigned to the correct cell clusters and only a few cells were misassigned.

Because there is no female or male bias in the Harmony integrated data, this suggests that the transcriptomes of male and female cells of each cell type are very similar to one another. To

further test this, male-specific and female-specific marker genes were called for each cell type cluster using the Seurat FindMarkers function. Primary pigment and cone cell clusters were excluded due to the small number of female cells present. Seurat called 689 potential sex-specific marker genes (Supplementary Data 7). FeaturePlot analyses of all potential sex-specific marker genes were generated and manually analyzed to check if they are

**Fig. 5 Pigment cell cluster identification and *santa-maria* and *wrapper* are two novel markers for adult pigment cells. a** Pigment cell clusters are colored in red in Seurat UMAP cluster plots of 7-day-old adult eyes. **b–e** FeaturePlots showing the expression of *w* (**b**), *Pdh* (**c**), *santa-maria* (**d**), and *wrapper* (red arrow, (**e**). All cells expressing *w*, *santa-maria* and *wrapper* were brought to the front in FeaturePlots (**b**), (**d**), and (**e**). **f** ViolinPlot of log-normalized *Pdh* expression in all cell types. **g** Tangential view of a 7-day-old adult eye prepared from a *santa-maria-T2A-Gal4 > UAS-mCherry-nls* animal and costained for mCherry and Elav. The green dotted hexagon marks the border of one ommatidium. **h** Coronal view of a 7-day-old adult eye from a *santa-maria-T2A-Gal4 > UAS-mCherry-nls* animal. **i** Coronal view of immunofluorescence images of 7-day-old adult eyes from a *wrapper-T2A-Gal4 > UAS-mCherry-nls* animal costained with Elav. **j**, **k** Tangential and coronal views of immunofluorescence images of 7-day-old adult *wrapper-T2A-Gal4 > UAS-mCherry-nls* eyes costained with Ct. Scale bars: 10 μm. **l** Schematic of an ommatidium with the red dotted boxes and black brackets marking the focal planes shown in (**g–k**).

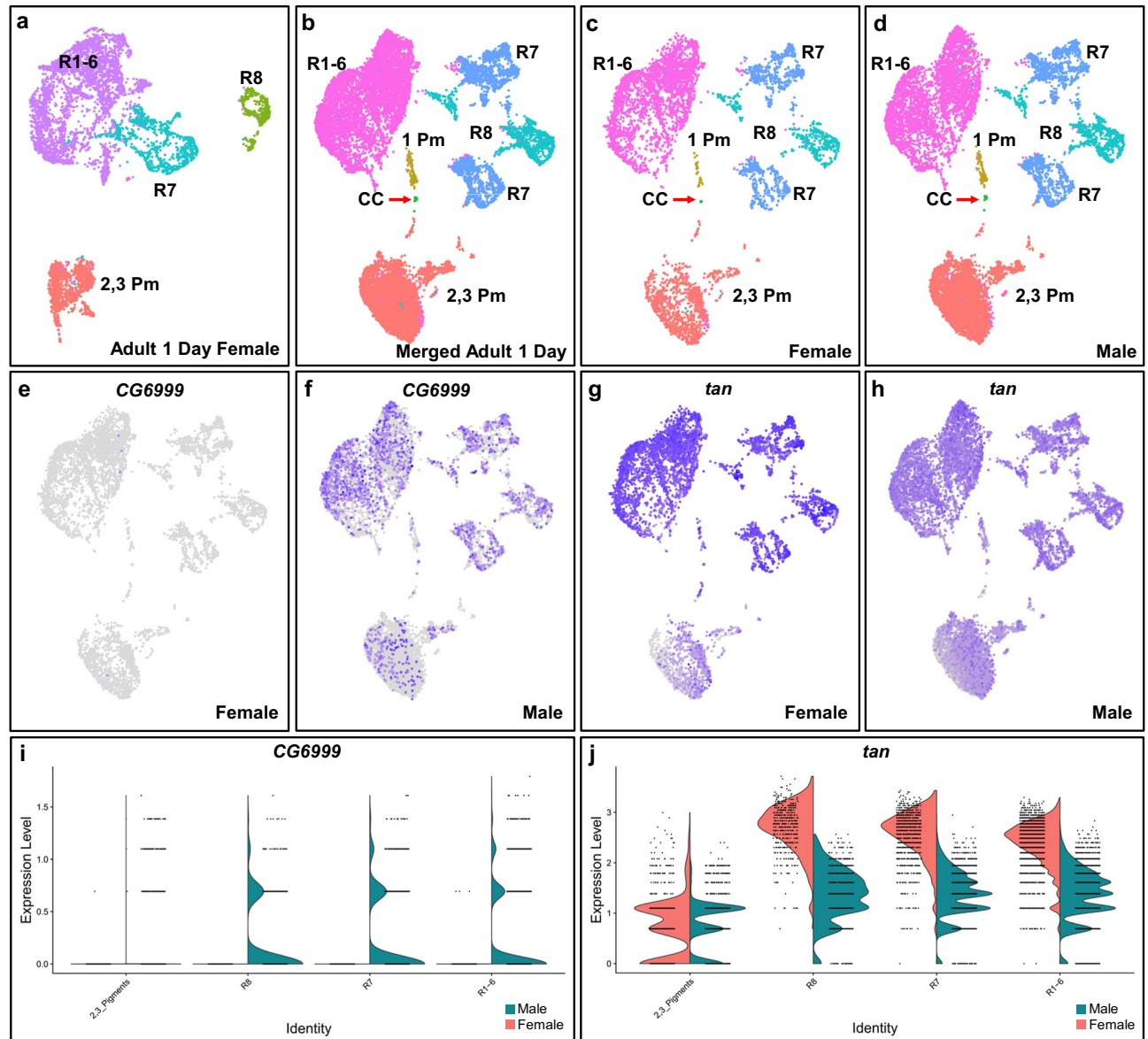

**Fig. 6 One-day-old adult male and female eyes have highly similar transcriptomes. a** Major cell type clusters are annotated on a Seurat UMAP cluster plot of 1-day-old adult female eye scRNA-seq. R1-6 (purple), R7 (blue), and R8 (green) label R1-8 photoreceptor clusters and PM (orange) labels pigment cell clusters. **b** Male and female adult eye scRNA-seq integrated into one Seurat UMAP cluster plot using Harmony. **c**, **d** Integrated male and female scRNA-seq cluster plot but split into female only (**c**) and male only (**d**) cluster plots. **e**, **f** FeaturePlots of *CG6999*, one of the two genes that show a male/female specific expression pattern. **g**, **h** FeaturePlots showing *tan* as an example of a gene that was called as a potential male or female-specific gene but shows no male or female specificity. All cells expressing *CG6999* and *tan* were brought to the front in FeaturePlots (**e–h**). **i**, **j** ViolinPlots showing log-normalized *CG6999* and *tan* expression in male and female cell types. Primary pigment and cone cells are omitted (see main text).

expressed in a sex-specific pattern. However, 687 of these potential sex-specific marker genes are expressed in both male and female cells with no clear bias for either sex. The only exceptions are *CG6999* and *apolipophorin* (*apolpp*), both of which

are expressed in most male cells but not in the vast majority of female cells (Fig. 6e, f, i and Supplementary Fig. 10). In addition, most of the called sex-specific genes are expressed at similar levels and in similar patterns between male and female cells. Only 21

**Table 3 Number of male or female cells by cell type in merged data set.**

| Cell type | 1-Day ♂ | 1-Day ♀ | Merged 1-Day | [a]Expected % |
|---|---|---|---|---|
| R1-6 | 4150 (37%) | 3064 (54%) | 7214 (43%) | 30% |
| R7 | 1739 (16%) | 997 (17%) | 2736 (16%) | 5% |
| R8 | 803 (7.2%) | 449 (7.8%) | 1252 (7.4%) | 5% |
| Cone cells | 32 (0.29%) | 6 (0.10%) | 38 (0.22%) | 20% |
| 1° Pigment | 112 (1.0% | 33 (0.58%) | 145 (0.86%) | 10% |
| 2°, 3° Pigment | 4328 (38.8%) | 1177 (21%) | 5505 (33%) | 20% |
| Total | 11,164 | 5726 | 16,890 | 90% |

[a]Expected % does not add up to 100% as bristle cells are not counted.

out of the remaining 687 potential sex-specific marker genes have greater than a 2-fold difference in average expression in male versus female cells in corresponding cell type clusters. An example is *tan*, which is expressed at higher levels in female photoreceptors (Fig. 6g, h, j). However, FeaturePlot analyses of *tan* shows expression patterns that are nearly identical between male and female cells (compare Figs. 6g and 6h). The remaining 20 genes also show nearly identical expression patterns between male and female cells. Therefore, most of the 689 potential sex-specific marker genes do not appear to be sex-specific (i.e., expressed in one sex but not in the other). In addition, although a few genes are expressed at different levels between the two sexes they still show the same expression patterns in the eye. In summary, UMAP clustering results suggest that the transcriptomes of male and female cells are highly similar for each major eye cell type and the absence of many sex-specific marker genes is consistent with this interpretation.

**R7- and R8-specific *Rhodopsins* are major determinants of R7 and R8 clustering.** The UMAP clustering of adult eye cells shows that R7 cells are split into two different clusters, where one expresses *Rh3* and the other expresses *Rh4* (Figs. 2i–k, 7a). Similarly, R8 cells are split into two groups, with one expressing *Rh5* and the other *Rh6* (Figs. 2a–c, 7a). R7 and R8 clustering appears to be strongly correlated to *Rh3-6* expression in all three time points (Fig. 7a, e, i, Supplementary Figs. 2 and 3). This observation suggests that *Rh3-6* may be major determinants in the separation of inner photoreceptor subclusters. To test this, we removed *Rh3* and *Rh4*, *Rh5*, and *Rh6*, or all four *Rhodopsins* prior to repeating our UMAP clustering. This removes the effects of *Rhodopsins* in the separation of R7 and R8 subtypes clusters. We anticipated that *Rh* removal will greatly impact cell clustering, and assigning specific photoreceptor identities will be difficult. We therefore transferred cell cluster identities from unaltered UMAP clustering to the *Rh*-removed cluster plots for each time point.

When *Rh3* and *Rh4* are excluded from the clustering process, the R7 clusters no longer form distinct *Rh3*-positive or *Rh4*-positive subclusters; *Rh3*- and *Rh4*-positive cells are mixed together (Fig. 7b, f, j). R7 cells are also partially embedded in the R1-6 cluster when *Rh3* and *Rh4* were removed. In contrast, the original clustering shows *Rh3* and *Rh4* cells form separate clusters and are separated from R1-6 cells (Fig. 7a, e, i). In addition, *Rh5* and *Rh6* expressing R8 cells remain separate from each other, which is also observed in the unaltered clustering. As expected, the pigment cells and cone cells clusters are unaffected by *Rh3* and *Rh4* removal in all three time points. These results show that *Rh3* and *Rh4* are largely responsible for the separation of *Rh3* and *Rh4* expressing R7 clusters for each time point. This suggests that *Rh3* and *Rh4* expressing R7s are transcriptionally similar and *Rh3* and *Rh4* expression are the primary transcriptional differences between these two types.

Similarly, when *Rh5* and *Rh6* are removed from clustering, the *Rh5* and *Rh6* subclusters are no longer distinct or separated from one another (Fig. 7c, g, k). *Rh5* and *Rh6* expressing R8s are mixed with one another in all three time points. A number of R8 cells are also embedded in the R1-6 cell clusters. In contrast, cluster plots without *Rhodopsin* removal show that *Rh5* and *Rh6* subclusters are distinct and separated from one another and from R1-6 clusters (Fig. 7a, e, i). *Rh3* and *Rh4* cell clusters remain distinct from each other and from the other photoreceptor clusters in all three time points. Similar to the unaltered cluster plot, the dorsal third R7 subcluster that expresses both *Rh3* and *Rh4* can be observed in *Rh5* and *Rh6* removed clustering (Supplementary Fig. 11, red arrows). These results show that *Rh5* and *Rh6* are crucial for the distinct clustering of *Rh5*- and *Rh6*-expressing R8 cell clusters but *Rh5* and *Rh6* do not play a major role in the separation of R7 cell clusters. This suggests that *Rh5*- and *Rh6*-expressing R8 cells are transcriptionally similar and *Rh5* and *Rh6* expression are the major differences between the two R8 types.

When *Rh3-6* are all removed from clustering, R7 and R8 cells are no longer in well-separated and distinct clusters in any of the three time points (Fig. 7d, h, l). Instead, *Rh3* and *Rh4* R7s form an intermixed cluster and mixed *Rh5* and *Rh6* R8s cluster directly adjacent to the R7 cluster. Similar to the *Rh3-4* removal or *Rh5-6* removal above, some R7s and R8s clustered into the R1-6 cluster when *Rh3-6* are removed. Again, pigment cell clusters are not affected when *Rh3-6* are removed. Notably, when *Rh3-6* are removed, R7 and R8 cells now cluster much more closely to the point that the two clusters are touching and they have an appearance of one loose cluster. At the boundary of R7 and R8 clusters, there is some intermixing of R7 and R8s. These results suggest that *Rh3-6* are also contributors to the separation of R7 and R8 cell clusters.

We have repeated the *Rh* removal experiment with *ninaE* to test if *ninaE* is a major contributor to R1-6 clustering (Supplementary Fig. 12). We found that removal of *ninaE* alone does not greatly change the clustering of R1-6 or any other cell clusters. However, R1-6 clusters now extend toward the secondary and tertiary pigment cell clusters in the 3-day and 7-day-old data sets (Supplementary Fig. 12b, h, red arrows). When *ninaE* and *Rh3-6* are all removed from clustering, R7 and R8 cells cluster closely together, similar to when *Rh3-6* are removed for 3-day and 7-day-old data sets (Supplementary Fig. 12f, l). In addition, R7 and R8 clusters are slightly closer to the R1-6 cluster when *ninaE* and *Rh3-6* are removed than when *Rh3-6* alone are removed. The R1-6 clusters also extend toward the pigment cell clusters in 3-day and 7-day-old data sets when *ninaE* and *Rh3-6* are removed (Supplementary Fig. 12f, l, red arrows). No drastic increase in photoreceptor intermixing is observed when *ninaE* and *Rh3-6* are removed. In the 1-day-old data set, removal of *ninaE* does not have any effect on cell clustering (Supplementary Fig. 12n). Removal of *ninaE* and *Rh3-6* does not markedly change

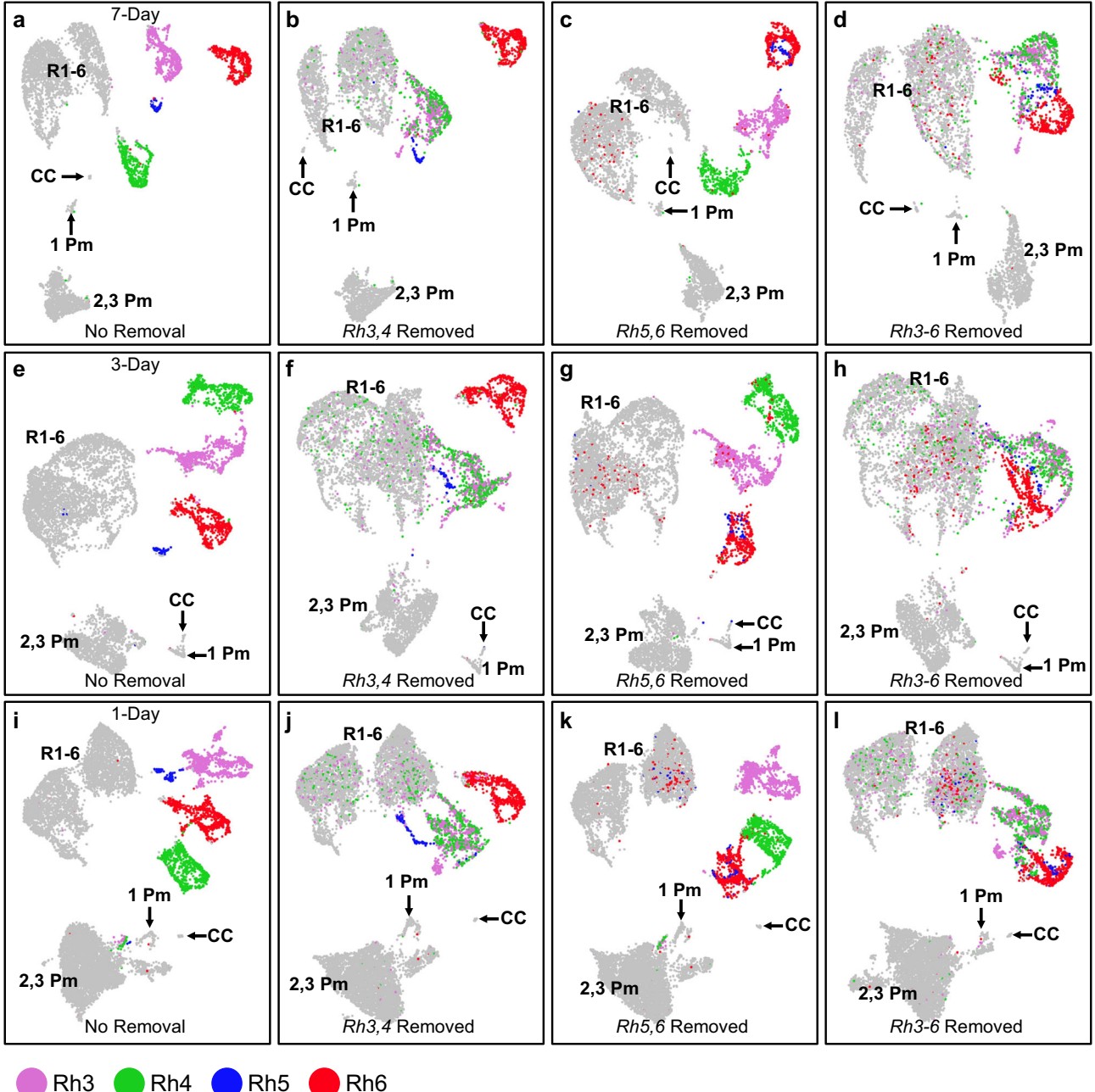

**Fig. 7 *Rh3*, *Rh4*, *Rh5*, and *Rh6* are major contributors to the clustering of R7 and R8 cells. a** UMAP clustering of 7-day-old adult eyes where R7 and R8 cells are shown as four distinct clusters expressing either *Rh3*, *Rh4*, *Rh5*, or *Rh6*. **b–d** UMAP clustering of 7-day-old adult eyes where *Rh3/4* counts data (**b**), *Rh5/6* counts data (**c**), or *Rh3-6* counts data (**d**) were removed from clustering. **e** UMAP clustering of 3-day-old adult eyes. **f–h** UMAP clustering of 3-day-old adult eyes where *Rh3/4* counts data (**f**), *Rh5/6* counts data (**g**), or *Rh3-6* counts data (**h**) were removed from clustering. **i** UMAP clustering of 1-day-old adult eyes. **j–l** UMAP clustering of 1-day-old adult eyes where *Rh3/4* counts data (**j**), *Rh5/6* counts data (**k**), or *Rh3-6* counts data (**l**) were removed from clustering. Only R7 and R8s were colored for clarity. Gray cell cluster identities were labeled: R1-6 are R1-6 photoreceptors; CC are cone cells; 1 Pm are primary pigment cells; 2,3 Pm are secondary and tertiary pigment cells.

the clustering in the 1-day-old data set compared with removal of *Rh3-6* alone (Supplementary Fig. 12r). These results suggest that *ninaE* is not a major contributor to the clustering of R1-6 cells.

**Vision and pigment production-related GO terms are enriched in photoreceptors and pigment cells marker genes, respectively.** We performed Gene Ontology (GO) term enrichment analyses of the cell type marker genes from each male time point with PANTHER (Fig. 8)[43]. We excluded primary pigment and cone

cells from GO analyses because they are underrepresented in all time points. Since there are extensive overlaps of marker genes for each cell type from all three time points, we pooled the marker genes for each photoreceptor subtype from all three time points and used that list for the GO term enrichment analyses. As expected, R1-6, R7, and R8 photoreceptors marker genes from 1D, 3D, and 7D adult eyes are enriched for GO terms related to vision or phototransduction (e.g., Rhodopsin signaling and light adaptation). Similarly, secondary and tertiary pigment cell marker genes from 1D, 3D, and 7D are enriched for GO terms related to

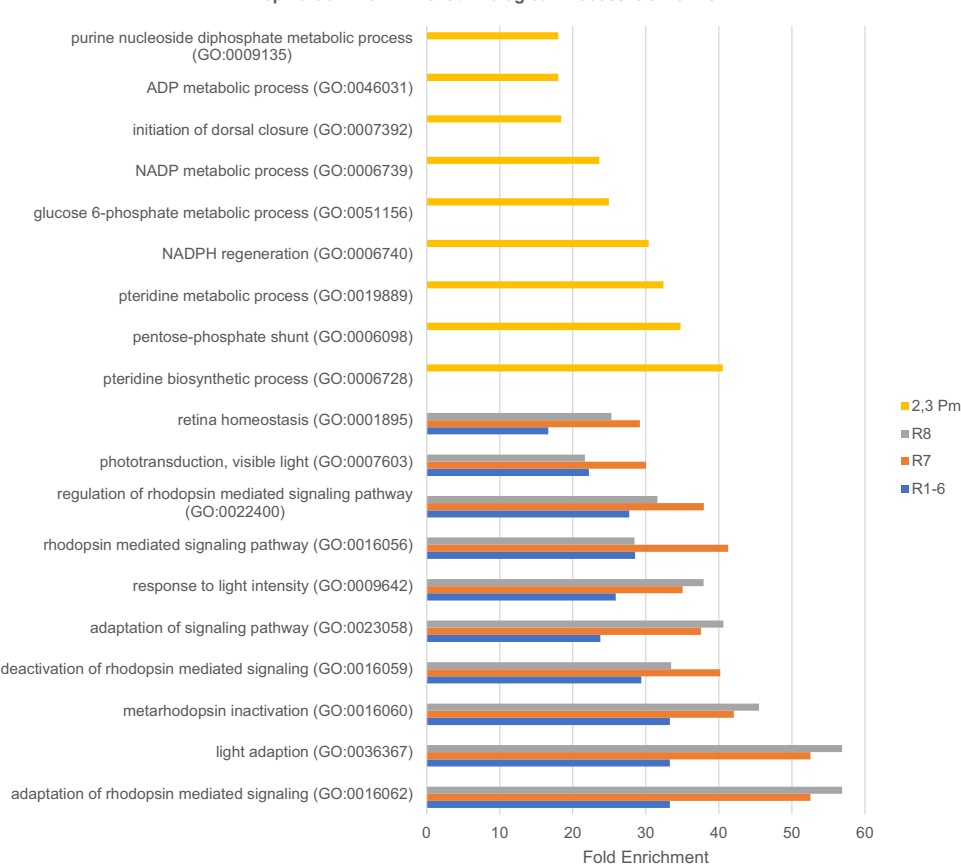

**Fig. 8 Top enriched Gene Ontology terms for photoreceptor and pigment cell marker genes are related to light reception and pigment production, respectively.** Bar graphs showing the fold enrichment of the top ten enriched Gene Ontology (GO) terms for pooled marker genes from 1-day, 3-day, and 7-day-old adult eyes for secondary and tertiary pigment cells (2,3 Pm) and R1-6, R7, and R8 photoreceptors.

processing of Purine or Pteridine, which are known precursors of the *Drosophila* eye pigments (Drosopterins)[44]. For each cell type, we compared the GO term enrichment between different time points but similar GO terms were enriched between each time point. This is consistent with the lack of aging transcriptome differences between the three time points as shown in the Monocle 3 analysis above.

We compared enriched GO terms of marker genes among photoreceptor subtypes (e.g., R1-6 vs R7) but this failed to show any significant differences in enriched GO terms. This suggests that the marker genes of each photoreceptor subtype have similar GO term enrichment. This observation is consistent with our Rhodopsin removal results above, which suggests that the major transcriptomic differences between R1-8 can be attributed to their Rhodopsin expression. Thus, it is not surprising that the GO terms of the photoreceptor subtype marker genes are all enriched in a similar fashion.

**Discussion**

In this report, we present single-cell resolution transcriptomic data of *Drosophila* adult eyes from 1-day, 3-day, and 7-day-old *Canton S* flies. scRNA-seq from the adult eye poses several technical challenges. First, the adult retina is firmly attached to the hydrophobic lens and cuticle. Physical removal of the lens without damaging the underlying retinal cells is extremely difficult. When left intact, the lens and cuticle is a solid barrier to chemicals and enzymes used for dissociation. Second, adult eye cells are elongated and adhere to each other in vivo and this

makes dissociating the adult eye into viable single cells much more challenging than many other tissues. Two of our data sets are from aged animals and aging is known to correlate with a decrease in the total number of transcripts per cell[25]. In fact, we observed a general decrease in the number of genes detected and number of reads from our 1-day-old to 7-day-old data sets (Supplementary Fig. 1c, d). The lower number of transcripts per cell poses a difficulty in single-cell RNA-sequencing where transcripts in each cell may not be captured, converted to cDNA and sequenced. This may lead to a loss of detected transcripts or loss of detected cells due to low transcript numbers. Despite these challenges, the quality control metrics our data sets indicate that the samples are of high quality (>80% viability and >600 median genes per cell) and our data sets captured over 6000 cells each. Our scRNA-seq data sets show that all major cell types in the eye are captured and different cell types form clear and distinct clusters. The clustering observed in our data sets accurately reflects the known cell types in the eye. For example, we observe that R7 and R8 photoreceptors are well separated and the pale and yellow R7 and R8 subtypes form their own distinct clusters. To our knowledge, such a high-resolution representation of the different cell types in the *Drosophila* adult eye has not been reported previously.

We have compared our data sets in two ways. First, we used Monocle 3 to find any pseudotime trajectories between the three male time points[23,24]. Our results revealed that the transcriptomes of 1-day to 7-day-old adult eyes do not change substantially, and no clear trajectories could be plotted. Second, we compared our 1-day-old adult male and female eye data sets. The

quality control metrics are similar between the two data sets. Harmony integration[42] reveals that the male and female data sets are highly similar and only two sex-specific genes were discovered in our data sets.

Although we have captured and sequenced all major cell types in the eye, including all photoreceptor subtypes, all pigment cell types and cone cells, we observed that cone cells and primary pigment cells are underrepresented in our data sets (Table 2). Cone cells and primary pigment cells are positioned most apically and are located next to the cuticle/lens of the eye. Since our dissociation protocol uses proteinase on adult eyes with the lenses intact, the cone cells and primary pigment cells would be the cell types that are least exposed to the proteinase. This may account, at least in part, for why we observe fewer cone and primary pigment cells dissociated into solution for sequencing when compared to the other cell types. We have tried increasing dissociation incubation time or adding mechanical dissociation steps to our protocol but both modifications result in a dramatic decrease in cell viability, leading to poor quality scRNA-seq results. We reason that high viability for most cells is of greater value than producing a dataset with good representation of all cell types but with severely compromised viability and quality. Despite primary pigment and cone cells being underrepresented, they still form their own clusters and can be identified in the Seurat UMAP clustering. We have validated that these are primary pigment cells using the novel primary pigment cell and cone cell marker genes, *wrapper* and *CG5597*, respectively.

The majority of the ommatidia in the eye can be classified as either pale or yellow based on Rhodopsin expression in R7 and R8 cells (Rh3-expressing R7s and Rh5-expressing R8s are pale; Rh4-expressing R7s and Rh6-expressing R8s are yellow)[27,28]. The ratio of pale to yellow ommatidia in an adult eye is about 30:70. However, the ratios of captured pale:yellow R7s and R8s are not in the expected ratios (10:90 to 13:87 for R8s and about 50:50 for R7 in our data sets, Supplementary Figs. 2f, 3f). While *Rh6*-expressing R8s are well represented in our data sets, the number of *Rh5*-expressing R8s are low and are consistently underrepresented. The number of pale or yellow R7s do not appear to be obviously underrepresented like pale R8s. Immunostaining of our *Canton-S* adult eyes show the expected ratios of pale to yellow R7s and R8s (Supplementary Fig. 2g); thus the discrepancy between the expected ratios and our results are not due to the input tissue. It is not likely that R7s and R8s changed their Rhodopsin expression to a different Rhodopsin during sample preparation as we have treated our sample with Actinomycin D, which inhibits all transcription. Therefore, we hypothesize that Rh5-expressing R8s may be preferentially lost during sample preparation or there may be a bias against *Rh5* expressing R8s in the data acquisition pipeline. It is possible that adult *Rh5* expressing R8 are especially sensitive to the dissociation conditions for scRNA-seq and thus are lost during sample preparation and filtered out. For R7s, we hypothesize that there may be a bias for capturing pale R7s. However, we cannot exclude the possibility that there may be a bias against yellow R7s in the pipeline.

Although a single nuclear RNA-seq (snRNA-seq) data set of whole adult *Drosophila* heads, which includes the eyes, was previously reported[45], our data presented here are single-cell RNA-seq data (scRNA-seq). The transcriptome data from snRNA-seq and scRNA-seq differ. First, because snRNA-seq are nuclear in origin, the extracted RNA molecules are biased toward unspliced versions. In contrast, scRNA-seq should not have a bias toward spliced or unspliced RNA. Bias for unspliced forms may cause complications in RNA velocity analyses which elucidate the dynamics of changes in transcriptome using spliced and unspliced transcripts[46]. Our established scRNA-seq protocol can also be adapted for a different, more sensitive scRNA-seq

technique such as SMART-seq to detect any cell type-specific splicing. Second, snRNA-seq data capture fewer RNA molecules per cell compared to scRNA-seq because each nucleus is smaller than a whole cell. Thus for aging tissues, where the total number of transcripts decrease, scRNA-seq offer an advantage over snRNA-seq. Third, studies comparing snRNA-seq and scRNA-seq in human liver and human brain tissues show that certain cell type-specific transcripts are only present in scRNA-seq but not in snRNA-seq and vice versa[47,48]. We compared our cell-type marker gene lists with the lists from published snRNA-seq[45]. Although there is overlap between the photoreceptor marker gene lists from the two data sets, *CG2082* and *santa-maria*, an R8 marker and secondary and tertiary pigment cell marker genes, respectively, identified and in vivo validated in this work were not identified as marker genes for R8 and pigment cells in the published snRNA-seq. Differences in marker gene lists between the two data sets may be expected since different methods were employed (single nucleus vs single cell). In summary, these overlaps and differences suggest that our scRNA-seq and the published snRNA-seq are complementary.

Seurat analyses and UMAP clustering of our data sets identified all photoreceptor subtypes and a substantial number of pigment cells in all time points. Each identified cell type cluster expresses known marker genes for the corresponding adult cell type. Interestingly, most previously characterized cell-type specific marker genes from larval eye discs are no longer expressed in the same specific cell types in the adult eye (Supplementary Table 1[36,49–72]). We validated the identities of clusters using T2A-Gal4 drivers of marker genes and immunostaining with antibodies recognizing known markers of specific cell types. Differential gene expression and FeaturePlot analyses of our data sets produce many potential novel markers for each identified cell type. GO-Term enrichment analyses on the marker lists of well-represented cell types show photoreceptors and pigment cells have enriched GO-terms relating to phototransduction and pigment production, respectively. To further test the validity of our cluster annotation and the specificity of cell type-specific markers, we validated several candidates in vivo, including *CG2082* for adult R8s, *igl* for both R7 and R8, *Zasp66* for adult photoreceptors, *CG5597* for cone cells, *wrapper* for primary pigment cells and *santa-maria* for secondary and tertiary pigment cells. In vivo validations of marker genes in this report all show specific expression patterns that closely match the expected expression pattern predicted from Seurat FeaturePlots. It is likely that most of the other potential marker genes identified in this study will also follow the expected expression pattern. Interestingly, the in vivo validated marker genes in this study are either uncharacterized (e.g., *CG2082*, *CG5597*) or their functions in the corresponding cell types are unclear (e.g., *wrapper* in primary pigment cells). Except for *santa-maria*, eye phenotypes have not been previously reported for the validated marker genes. Further studies of these marker genes may lead to new insights in the maintenance or function of the adult eye.

Our scRNA-seq data sets have identified numerous novel marker genes for each major cell type in the adult eye. Prior to this study, cell type-specific markers for the adult eye were very limited. The novel marker genes identified here address the lack of cell type-specific markers for this stage. The cell type-specific marker genes also provide novel targets for constructing new reagents and genetic tools (e.g., for driving transgenes). Many marker genes are conserved in humans and the conserved human homologs have associated disease phenotypes. These marker genes are likely to benefit research that uses the adult eye as a model system. The genes discovered by our data sets may play key roles in the function or maintenance of the adult eye and thus may play key roles in phototransduction and neurodegeneration.

Interestingly, we noticed that the T2A-Gal4 driven reporters often show a more specific expression pattern than suggested by the FeaturePlots. FeaturePlots show relative expression of selected genes but not the absolute expression[20]. Thus, for genes that are expressed weakly, any fluctuation in expression levels may be exaggerated. We used SoupX to remove ambient RNA from our data sets but it may not remove all ambient RNA (e.g., *ninaE*)[19]. Thus the FeaturePlots may show cells with unexpected expression of cell type markers simply due to the presence of ambient RNA sequenced in that cell. Despite these caveats, our FeaturePlots still show a clear enrichment of cell type markers in the expected cell clusters and our in vivo staining results show T2A-Gal4 driven reporters are expressed only in the expected cell type as suggested by the FeaturePlots. For example, *CG2082* > reporter is expressed in R8 only, which matches the *CG2082* expression enrichment in R8s as shown in the FeaturePlots. Although some *CG2082* positive cells are seen in R1-6 in the FeaturePlots, no reporter expression in R1-6 was observed in the adult eye. This suggests that *CG2082* is not strongly expressed or has no expression in R1-6 cells but *CG2082* is strongly expressed in R8 cells. Taken together, the FeaturePlots are accurate predictors of the in vivo expression patterns of marker genes in cell type clusters where the expression is enriched.

Our scRNA-seq results show that *ninaE* expression is not restricted to R1-6 cells as expected (Fig. 3b) but it is also expressed to a lower degree in all other clusters. In contrast, the other *Rhodopsins* (*Rh3-6*) do not show nearly as much expression in unexpected cell types. There are at least two hypotheses that may explain this observation. First, all eye cells may express *ninaE* transcripts but NinaE protein is translated only in R1-6. Second, *ninaE* transcripts are expressed only by R1-6 and the *ninaE* transcripts detected in non-R1-6 cells represent ambient RNA. Of the two hypotheses, we favor the second for two reasons. First, R1-6 account for six different cells per ommatidium. This outnumbers all other cell types per ommatidium. Second, NinaE proteins are one of the most highly abundant proteins in the adult eye[73]. This was expected as NinaE proteins are localized to the entire R1-6 rhabdomeres and R1-6 rhabdomeres are large and span nearly the entire depth of the retina[74,75]. Therefore, *ninaE* transcripts are expected to be highly abundant in R1-6. Indeed, all of our datasets show that *ninaE* is the most abundant transcript. Since *ninaE* is the most highly expressed gene in the eye, it is predicted to contribute to ambient RNA more than any other gene.

In the adult eye, R1-6 photoreceptors are needed for motion detection while pale and yellow R7s and R8s are needed for color detection[12,13]. Pale and yellow R7 and R8s are used to detect different colors and UV light (Rh3: ~345 nm UV light, Rh4: ~375 nm UV light, Rh5: green, Rh6: blue)[76,77]. In all time points, Seurat clustering segregated R1-6 as one cluster that is distinct from R7s and R8s, which form their own distinct clusters. R7s and R8s are further separated into clusters that represent their pale/yellow subtypes. It is thought that specific wavelengths of light activate the expressed Rhodopsins in each photoreceptor, which then activates a downstream phototransduction cascade that is common to all photoreceptors[14,78]. One implication of these observations is that all photoreceptor subtypes may be transcriptionally similar and which Rhodopsin is expressed may be the main difference between subtypes. Consistent with this model, Seurat clustering strongly reflects the type of opsin expressed by each photoreceptor subtype at all time points.

To test this model, we removed the effects of R7 and R8 specific *Rhodopsins*, *Rh3-6*, in the clustering process. Removal of *Rh3* and *Rh4* causes the R7 clusters to collapse and *Rh3* and *Rh4* R7s are intermixed with one another. Similarly, when *Rh5* and *Rh6* are removed, all R8s cells are intermixed. When *Rh3* to *Rh6* are removed, R7 and R8 clusters collapse into one loose cluster. These results suggest that the major drivers of R7 and R8 clustering are the R7 and R8 specific *Rhodopsins* and apart from *Rhodopsin* expression, R7 and R8 cells are very similar transcriptionally. This implies that other than *Rh3-6*, there should be very few markers that distinguishes R7 and R8. Indeed, our differential gene and FeaturePlot analyses of R7 and R8 marker genes identified very few R7 or R8 specific marker genes; instead, the majority of novel markers for R7 and R8 identified in this work are expressed in both subtypes. Interestingly, we noticed that some but not all R7 and R8 cells are mixed with R1-6 clusters after *Rh3-6* are removed. This also suggests that *Rh3-6* are determinants in separating R7 and R8 from R1-6. But the lack of a complete intermixing of R7, R8, and R1-6 when *Rh3-6* are removed suggests other genes also contribute to photoreceptor clustering.

Although R1-6 cells form a distinct cluster and Seurat can call marker genes for the R1-6 clusters for all time points, FeaturePlot analyses showed these markers are not specific to R1-6; instead, most are specific to all photoreceptors. We observed that the R1-6 clusters have *ninaE* as their top marker while *Rh3-6* were not called as markers in the R1-6 clusters in all time points. Therefore, we predict that if *ninaE* was not a major ambient RNA, it would be a definitive R1-6 marker gene in our FeaturePlots. However, *ninaE* is not a major contributor to R1-6 clustering as the removal of *ninaE* does not change the clustering of R1-6 clusters. Other contributors (e.g., R8-specific and R7/8-specific markers) may also contribute to defining the R1-6, R7, and R8 photoreceptor clusters. In the case of R7 and R8, our *Rh* removal results suggest that R7 and R8 are transcriptionally similar to one another and this may account for the lack of R7-specific markers. We hypothesize that R1-6 cell clusters are also somewhat transcriptionally similar to R7 and R8. Thus, R1-6 specific marker genes may be rare and may not be detected in our data sets, similar to the lack of R7-specific markers.

In summary, our work presents transcriptomic data of *Drosophila* adult eyes prepared from whole cells at a single-cell resolution. Our data sets show that there are very few sex-specific differences between male and female adult eyes. Analyses of our data sets show that *Rhodopsin* expression is a major contributor to the transcriptomic differences between photoreceptor subtypes. Finally, we identified numerous marker genes with differential expression analyses for all the major cell types in the adult fly eye and we have verified novel markers for each major cell type in the adult eye in vivo. The function of many of the cell type-specific marker genes are unknown but they may perform critical roles in the maintenance and function of the adult eye. In addition, many of the marker genes are conserved and have been associated with human diseases. These marker genes will also serve as valuable tools for generating cell type-specific reagents for field.

## Methods

**Adult eye dissociation and scRNA-seq.** Forty adult eyes were dissected into 500 µL Rinaldini solution with 1.9 µM Actinomycin D (Sigma-Aldrich, A-1410) in a 2 mL LoBind Eppendorf tube on ice. Once all eyes were dissected, 25 µL of 100 mg/mL Collagenase (Sigma-Aldrich, C-9697) was added to start the dissociation. The tube was placed flat onto a horizontal shaker and shaken at 250 rpm at 32 °C. The adult eyes and the dissociation solution was gently pipetted up and down 30 times every 10 min with a P1000 tip. The degree of cell dissociation was monitored after each pipetting step by viewing 2 µL aliquots of dissociating eyes with brightfield microscopy. Incubation and pipetting were repeated until only single cells were observed in the solution; usually this takes ~25–30 min. Dissociation was stopped by adding 500 µL of ice-cold Rinaldini solution to the dissociated eyes. An 8 µm filter, prewet with ice-cold Rinaldini solution, was used to filter dissociated single eye cells and to remove any cuticle and undissociated cell clumps. Single eye cells were allowed to drain by gravity into a new 2 mL LoBind Eppendorf tube. The filter was washed once with 1 mL of ice-cold Rinaldini solution with 1% BSA and the wash through was pooled with the single-cell solution. Single cells were pelleted by centrifugation at 50 × g for 5 min at 4 °C. Supernatant was discarded and single cells were very gently resuspended with a

wide bore P1000 pipette tip in 1 mL ice-cold Rinaldini solution with 1% BSA. Single cells were pelleted again by centrifugation at $50 \times g$ for 5 min at 4 °C. Washed and pelleted single cells were then resuspended in ~50 μL of Rinaldini solution with 1% BSA. Cell viability was assayed by staining a 2–5 μL aliquot of washed dissociated single cells with Hoechst-PI dye and cell concentration was assayed with the same aliquot. Samples with higher than 95% viability and a concentration of ~1000 cells/μL were chosen to proceed to cDNA library construction. Single-cell libraries were prepared according to Chromium Next GEM Single Cell 3' Reagent Kit v3.1 kit (10x Genomics). Briefly, single cells, reverse transcription (RT) reagents, Gel Beads containing barcoded oligonucleotides and oil were loaded on a 10x Genomics Chromium Controller to generate single-cell Gel Beads-In-Emulsions (GEMS). Full-length cDNA was synthesized and barcoded for each single cell. GEMs were then broken and cDNAs from each single cell were pooled. Following clean up using Dynabeads MyOne Silane beads, cDNAs were amplified by PCR. Amplified product was fragmented to optimal size before end-repair, A-tailing and adapter ligation. Final cDNA libraries were generated by amplification. KAPA Library Quantification kit (Roche) was used to quantify libraries. Libraries were sequenced with NovaSeq 6000 (Illumina) to a sequencing depth of ~100 million and ~500 million reads for 1-day-old male and ~500 million reads for 1-day-old female eyes, and ~100 million reads for 3-day and 7-day-old male eyes. FASTQ files generated from sequencing were analyzed with 10x Genomics Cell Ranger v6.0.1 using *Drosophila melanogaster* reference genome release 6 (dm6) as the reference genome.

**Seurat analyses**. Detailed R codes are available at (https://github.com/kygithubtokenaccount/Adult-Eye-scRNA-seq-R-codes/). Cell Ranger output was directly analyzed with SoupX v1.6.1 with default parameters to remove contaminating ambient RNA[19]. Cluster data input for SoupX was generated from the Cell Ranger output using Seurat v4.1.1 SCTransform workflow with default parameters and without any cutoffs[20,22,79]. The ambient RNA corrected SoupX gene matrix outputs were then analyzed with Seurat SCTransform workflow. Potential multi-cell containing droplets and unhealthy cells were filtered out using the following parameters: (1) only cells with mitochondrial mapping percentage <20% for 1-day (1D) male, <25% for 3-day (3D), <18% for 7-day (7D), and <25% for 1-day (1D) female were kept; (2) only cells with a total number of genes between 200 and 3500 for 1D male, between 200 and 2800 for 3D, between 200 and 2700 for 7D and between 200 and 4000 for 1D female were kept. Filtered cells were analyzed with SCTransform with some parameter changes: variable features = 3500, 2800, 2700, and 4000 for 1D male, 3D male, 7D male, and 1D female data sets, respectively. The percentage of all expressed genes that mapped to mitochondrial genes were regressed for each data set. RunUMAP and FindNeighbors functions were run at 50 dimensions for all datasets. Non-eye cells were identified by *found in neuron* (*fne*, a brain neuron marker), *moody*, and *reversed polarity* (*repo*, glial markers) expression and they were removed such that all data sets consist of only eye cells[80–82]. Raw counts from eye only data sets were extracted and Seurat SCTransform workflow was performed again to generate Seurat analyzed 1D-, 3D-, 7D-old male eye data sets and 1D-old female eye data set. Cell clusters were annotated with their cell type identity using FeaturePlots of known cell type markers (e.g., *Rh5* and *Rh6* for R8 photoreceptors). Marker genes for each cluster and each time point were identified using the FindAllMarkers function in Seurat with the following parameters: only.pos = T, min.pct = 0.25 and logfc.threshold = 0.25.

SoupX-treated raw counts from 1D male and female eye datasets were merged and normalized with SCTransform. Following this, Harmony was performed using default parameters to integrate the two data sets[42]. RunUMAP and FindNeighbors functions were performed with 50 dimensions. Clusters were annotated using FeaturePlots of known cell type markers. FindMarkers function was used to find marker genes that are male or female-specific for each cell cluster.

To generate the datasets for *Rh* removal, SoupX-treated raw counts were extracted from 1D, 3D, and 7D-old male eye data sets. Counts data of *Rh3*/4, *Rh5*/6, and *Rh3-6* were removed from the SoupX-treated raw counts. SCTransform was rerun on the modified raw count data for each dataset with the same parameters as described above. Cluster identities were transferred from the unaltered data to *Rh* removed data in Seurat.

**Monocle 3 analyses**. SoupX-treated raw counts from 1D, 3D, and 7D male eye datasets generated from Seurat were extracted. They were converted into the Cell Data Set (CDS) format using as.cell_data_set function within Seurat-Wrapper v0.3.0[20]. The CDSs were combined and analyzed with default parameters in Monocle 3 v1.0.0[23,24,83]. Batch effects correction was done with Batchelor v1.10.0 within Monocle 3[83]. Cell clusters were initially assigned by Monocle 3 with a resolution parameter of 1e-3 and their identities were annotated using cell type marker genes discovered in Seurat analyses above. Pseudotime trajectory was calculated within Monocle 3.

**GO term analysis**. Marker genes for each cell type were pooled from all three time points and then analyzed with Panther[43]. The fold enrichment of the top ten enriched GO terms for biological processes were graphed.

**Fly husbandry**. All flies were raised at 25 °C on cornmeal agar food as per standard protocols. The following fly stocks were obtained from the Bloomington Stock Center: *UAS-mCherry-nls* (38424), *UAS-mCD8-GFP* (5137), *CG2082-T2A-Gal4* (76181), *igl-T2A-Gal4* (76744), *Zasp66-T2A-Gal4* (93472), *CG5597-T2A-Gal4* (93309), *santa-maria-T2A-Gal4* (80598), and *wrapper-T2A-Gal4* (93483). *T2A-Gal4* lines were crossed with *mCherry* or *mCD8-GFP* reporter lines and progeny carrying both the Gal4 driver and the reporter were dissected for immuno-fluorescence imaging.

**Immunohistochemistry**. Freshly eclosed adult animals were aged for 7 days at 25 °C before dissection for immunofluorescence imaging. Adult eye dissection and staining were adapted from Hsiao et al.[84]. Adult eyes were dissected in PBS and were fixed in 3.7% paraformaldehyde in PBS for 30 min at room temperature. PBS + 0.3% Triton X-100 was used as PBST in all washes with PBST. Primary antibody incubation was done overnight at room temperature. Secondary antibodies incubation was changed to 1 h at room temperature instead of overnight at room temperature. Stained and washed adult eyes were mounted on bridged slides with lens side up for tangential images and lens side down for coronal images. Optically stacked images were taken on a Zeiss Apotome Imager microscope and processed with Zen Blue and Photoshop software. The following primary antibodies were used: rat anti-Elav (RRID:AB_528218, 1/1000), mouse anti-Pros (RRID:AB_528440, 1/100), mouse anti-Ct (RRID:AB_528186, 1/100), chicken anti-GFP (RRID:AB_300798, 1/1000), rabbit anti-mCherry (RRID:AB_2889995, 1/2000), mouse anti-Rh3 (gift from Dr. Steven Britt, clone 2E1 1/100), mouse anti-Rh5 (gift from Dr. Steven Britt, clone 7F1, 1/10), guinea pig anti-Rh6 (gift from Dr. Claude Desplan, 1/1000). Secondary antibodies were used at a concentration of 1/500: Cy5 anti-rat (RRID:AB_2340672), Cy5 anti-guinea pig (RRID:AB_2340460), Alexa 488 anti-rat (RRID:AB_141709), Alexa 647 anti-mouse (RRID:AB_162542), Alexa 488 anti-chicken (RRID:AB_2762843), Alexa 568 anti-rabbit (RRID:AB_2534017), Alexa 546 anti-rabbit (RRID:AB_2534016).

**Adult eye sectioning and imaging**. Adult eyes were fixed, embedded, and sectioned as per standard protocol[85]. Briefly, adult heads were roughly bisected and fixed in 0.1 M sodium phosphate + 2% glutaraldehyde + 1% $OsO_4$ and 2% $OsO_4$. Fixed eyes were dehydrated with 30%, 50%, 70%, 90%, 100%, and 100% ethanol solutions (10-min incubation each). Dehydrated adult heads were embedded in Durcupan Resin. The embedded eyes in resin blocks were entirely exposed using a razor blade and the exposed eye was sectioned directly. Eye sections were imaged with brightfield microscopy using a Zeiss Apotome Imager microscope. Images were processed in Photoshop. Optically stacked images of external adult eyes were taken with a Zeiss Apotome Imager microscope or a Zeiss LSM800 Confocal microscope. Images were processed in Zen Blue and Photoshop software.

**Statistics and reproducibility**. Each scRNA-seq sample was prepared with 40 eyes from at least 20 animals. More than 5000 filtered eye cells were obtained for each time point. The numbers of cells in each major cell type cluster (replicate transcriptome for each cell type) are summarized in Table 2.

**Reporting summary**. Further information on research design is available in the Nature Portfolio Reporting Summary linked to this article.

## Data availability
All raw and processed single-cell RNA-seq data were uploaded onto Gene Expression Omnibus, Accession number: GSE214510). Source data for GO-Term enrichment analyses are available in Supplementary Data 8. All other data (e.g., RDS files) are available upon request.

## Code availability
All R scripts used to generate the data shown in this work will be uploaded onto GitHub (https://github.com/kygithubtokenaccount/Adult-Eye-scRNA-seq-R-codes/).

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

## Acknowledgements
We thank Hugo Bellen, Steven Britt, Claude Desplan, Shinya Yamamoto for sharing antibody reagents and fly stocks. This work is funded by The Retinal Research Foundation. This work was performed at the Single Cell Genomics Core at Baylor College of Medicine partially supported by NIH shared instrument grants (S10OD023469, S10OD025240), P30EY002520 and CPRIT grant RP200504.

## Author contributions
K.Y. and K.K.B.R. performed the adult eye dissection and analyses of scRNA-seq data. Single-cell dissociation protocol was performed by K.K.B.R. Y.L. performed the cDNA libraries construction and sequencing of single-cell cDNA libraries. K.Y. prepared and conducted the immunofluorescence imaging of adult eyes with help contributed by Y.S. Manuscript was prepared by K.Y. and reviewed by K.K.B.R., R.C., and G.M.

## Competing interests
The authors declare no competing interests.
