## [Peer Review File · Communications Biology]

Reviewer #1

This is a thorough and expansive study single-cell transcriptomic analysis of 1,3,7 day old female and 1 day old male adult *Drosophila* eye providing results deriving from 27000 ommatidia cells, covering early and mature developmental states of the adult eye. They identify thus a number of novel marker genes for differential cell types in this tissue. The *Drosophila* eye has, and continues to be, a highly utilized and broadly used model system to monitor and pathway gene activity with important extensions in pathobiology. In addition their analysis suggests several hypotheses and avenues for further study regarding the mechanisms behind phototransduction itself.

Given the utility of the adult fly eye as a model system not only for studying phototransduction but also neurodegeneration and a broad variety of diseases, this paper represents a significant resource that will be of use in many future studies. The study is of course not mechanistic but it will be highly informative and thus valuable to a broad spectrum of researchers.

The study is well presented and experimentally well done and I recommend publication. Here are a few suggestions which we believe could improve the paper if addressed:

1. The R1-6 cluster (Figure 2) appears to consist of two lobes or subclusters. Is there any significance to this? Is there a difference in gene expression between these two subclusters, or is it merely an artifact of the clustering or visualization method?

We agree with the reviewer that R1-6 photoreceptors appear as 2 subclusters. This is more obvious in the 1-day and 7-day old datasets than in the 3-day old dataset.

We first speculated that the R1-6 subclusters reflect the dorsal/ventral positioning of the R1-6 photoreceptors. However marker genes for the dorsal half of the eye, *mirr*, *caup* and *ara*, are expressed by both R1-6 subclusters. This suggests that the two subclusters are not separated due to dorsal and ventral R1-6 cells.

We used the FindMarkers function in Seurat to identify marker genes that distinguishes the two R1-6 subclusters from each other. We then used Seurat FeaturePlots to screen the 261 marker genes. We only found 65 genes that are specific to photoreceptors and has some biased expression in one of the two R1-6 subclusters. Interestingly, we did not find any gene that is expressed in only one R1-6 subcluster. However, all of the 65 PR-specific markers are expressed in more cells and at higher levels in one subcluster versus the other. Following this, we checked two quality control metrics between the two R1-6 subclusters: the number of transcripts in each cell (nCount_RNA) and the number of genes detected in each cell (nFeature_RNA). We found that one of the subclusters has higher nCount_RNA and nFeature_RNA. Interestingly, the subcluster that has more nCount_RNA and nFeature_RNA also shows increased expression of the 65 R1-6 subcluster markers identified above. This suggests that the increased expression of R1-6 subcluster markers may be due to the increase in overall sequenced transcripts in the subcluster cells. It is unclear why some R1-6 cells have increased nCount_RNA and nFeature_RNA and the biological significance of this observation is unknown. Therefore, we would prefer to exclude this discussion from the manuscript.

2. In Figure 4, there appears to be a second group of cells expressing *ct*, *Crys*, and *CG5597* in the primary pigment cell cluster. Are these also cone cells that have been mis-attributed to the pigment cell cluster, or a bona fide group of pigment cells expressing these genes?

We thank the reviewer for this observation. In all our data sets, clustering was performed using Seurat, which is an unsupervised method to cluster cells based on the expression of all genes in each cell. We only assign cell type identities after clustering was finished. We also note that we have captured very few primary pigment cells and cone cells in our data sets. The low number of cone and primary pigment cells may affect the clustering accuracy of Seurat. As the reviewer suggested, it is possible that some cone cells were misplaced into the primary pigment cell clusters. We generated violin plots showing *ct* and *CG5597* expression in all cone cells and primary pigment cell (see below) and we observed that there were only 6/48 primary pigment cells that express *ct* and/or *CG5597* (red circled dots). Therefore the number of potentially misplaced cone cells are low and thus most of the cells in these two clusters should be assigned correctly.

Although we have powerful computation tools to identify cell type specific marker genes, we believe that *in vivo* validation of the identified marker genes is still crucial. In our *in vivo* verification of cone cell markers, we showed that *Ct* and *CG5597* are cone cell specific markers and we do not observe their expression in any pigment cells. Thus, those *ct* and *CG5597* expressing cells in the primary pigment cell clusters should not be primary pigment cells that express cone cell markers. We agree with the reviewer's suggestion that these 6 cells are likely to be cone cells that were misplaced into the primary pigment cell cluster. Since there were very few misplaced cells and the total number of captured cone cells and primary pigment cells were small in our time points, we would prefer to not discuss these cells in the manuscript.

We should note that our *in vivo* validation also shows the cone cell and primary pigment markers identified in this study are indeed cone cell and primary pigment cell specific, respectively. These observations support the accuracy of the marker genes elucidated for cone cell and primary pigment cells.

Violin plots showing expression of *ct* (left) and *CG5597* (right) in primary pigment cells (1_Pigment) and cone cells (Cone). Dots circled in red mark the *ct*- and *CG5597*-expressing cells that are assigned in the primary pigment cell cluster.

3. The authors find that for the most part there is no difference in gene expression in the eye between males and females (Figure 6). Are genes on the Y chromosome therefore not expressed in any of the eye tissues examined here?

We checked the FeaturePlots and Violin Plots of all genes on the Y chromosome in our datasets (1-day male and female, 3-day male, 7-day male) and none were significantly expressed in the eye tissues (only 2 genes show expression in 1-3 cells). To verify that there

are indeed no or very low expression of Y chromosome genes, we extracted the raw counts of all Y chromosome genes and checked for any expression. In all data sets, most Y chromosome genes have 0 transcripts detected and only two genes have 1 to 2 total detected transcripts. Therefore genes on the Y chromosome were not detected in any of the eye tissues examined in this study. Since this is a negative result, we would prefer to not include this observation in the manuscript.

4. I found it very interesting and somewhat surprising that no R1-6-specific markers were identified, other than *ninaE*, which has been well-established in the past as an R1-6-specific rhodopsin. However, *ninaE* does not appear entirely R1-6-specific in the current study. The authors reasonably speculate that this may be an artifact caused by presumptive ambient *ninaE* RNA. An analogous experiment to the Rh3/Rh4 or Rh5/Rh6 exclusion experiments in which *ninaE* is excluded from the clustering should be performed to address this issue. If *ninaE* is indeed a primary determinant of R1-6 identity, then I would expect for the R1-6 cluster to be redistributed while the remaining clusters are unaffected. This experiment would also provide support for the very intriguing hypothesis extended in the discussion that R1-6 cells are similar to R7 with the exception of expressing high *ninaE*.

We thank the reviewer for the interest in the photoreceptor clustering and the helpful suggestions. We have shown that removal of *Rh3-6* does not completely collapse the R7 and R8 clusters into R1-6. In the discussion, we reasoned that this observation suggests that there are other contributing factors (other than *Rh3-6*) that separate R7 and R8 from R1-6. However, we have not explicitly stated that some of the likely contributing factors may be the 24 non-*Rh3-6* R7 and R8 specific marker genes. An example is *igl*, which was *in vivo* validated as a R7+R8 marker gene in this manuscript and it is not expressed in R1-6. We have now included this in the discussion.

We have repeated the *Rh* removal experiment with *ninaE* removed and *ninaE* + *Rh3-6* removed. When only *ninaE* was removed, the R1-6 clusters do not completely collapse into the other cell type clusters. R1-6 cell clusters extend toward, but do not mix with, the pigment cell clusters for 3-day and 7-day old samples. There is little effect in the 1-day old data set. R1-6 cells also do not intermix with R7 and R8 cells.

When *ninaE* + *Rh3-6* are removed we observed that the R1-6 cluster appears to be more spread out and extends toward the pigment cell clusters when compared with the *Rh3-6* removed Cluster Plot for 3-day and 7-day old data sets. Some R7 and R8 cells are also positioned closer to the R1-6 clusters. We did not observe any clear additional intermixing of R1-6 and R7 and R8 cells. The additional removal of *ninaE* does not change the clustering of the 1-day old data set.

These results suggest that for 3-day and 7-day old data sets, *ninaE* is not a major contributor to R1-6 clustering away from R7 and R8 cells. *ninaE* has some contribution in distinguishing R1-6 cells from pigment cells in the 3-day and 7-day old data sets. Due to these observations, we were careful in not naming *ninaE* as a contributor to R1-6 clustering. To clarify these observations in the discussion, we have included the *ninaE* removal experiments in supplemental figures and changed the wording in the manuscript.

5. A great deal is known about the genes and pathways required for the specification of particular photoreceptor identities during *Drosophila* larval development, and at least some of them are known to continue to be expressed in adult eyes. Where do these

genes (e.g. *salm*, *svp*, *glass*, *spitz*, *Notch*, etc) map in the adult photoreceptors, and can they continue to be used as cell-type specific markers in the adult?

We thank the reviewer for this question. We have previously generated Featureplots against commonly used marker genes for cell types in L3 eye discs: *ato*, *sens* and *boss* for R8, *ro* for R2/5, *sev* and *svp* for R3/4, *B-H1* and *B-H2* R1/6, *sev* and *pros* for R7, *salm* and *salr* for R7+8, *ct* for cone cells. We discovered that most of these genes are no longer expressed in the adult eye. Our data shows that *salm*, *salr*, *sens*, *ct* and *pros* are still expressed in the same cell types in the adult eye and these results match with published antibody staining data. *boss* expression is no longer restricted to R8 and is strongly expressed in all photoreceptors in 1-day old adult eyes with weak expression in non-photoreceptor cells; *boss* is expressed in all photoreceptors in the 3-day and 7-day old data sets. The *boss* expression results match previously published data showing that *boss* is expressed in all photoreceptors during pupal development.

We also observed that major signaling pathway components do not follow the expression patterns seen in larval eye discs. For example, although *hh* is expressed by R1-7 in eye discs, *hh* is not expressed in any cells in the adult eye. Similarly, *wg* is not expressed in any cells in the adult eye. Although *spi* is expressed in some cells in all adult eye cell types, *pnt* is expressed in a few cells in the adult eye. *N*, *Ser* and *Dl* are not expressed in any adult photoreceptors. *gl* is expressed by all cells in the adult eye.

These observations show that most larval eye cell marker genes cannot be used as adult eye markers. Thus a new set of marker genes are needed to mark adult eye cell types. We have included a comment in the discussion about this observation and have also included a Supplemental Table of known larval cell type-specific genes and their expression in the adult eye in the manuscript.

6. How do the current findings compare to those from the previously published snRNA-seq dataset from adult heads mentioned in the discussion (reference 44)? Although the current dataset is likely more specific and high-resolution, I would still expect to see some overlap with the eye markers identified in that study, at the very least with the rhodopsin genes. Beyond that, do any of the novel markers identified in this study show any specificity in the snRNA-seq dataset?

We thank the reviewer for this question. We have obtained the marker gene lists from the R1-6, R7 and R8 photoreceptor clusters, cone cell and pigment cell clusters from the previously published adult fly snRNA-seq dataset (Li et al., 2022). These data were derived from 5-day old adult heads, while our data sets were from 1-, 3- and 7-day old animals. Although we could not perform a direct age matched comparison, we compared the published marker gene lists with our three sets of marker gene lists. In addition, the published marker lists include genes that were down-regulated. We have filtered out the down-regulated genes and any genes with <0.25 log fold-change to match our marker gene list parameters prior to comparing the marker lists.

We found that there is considerable overlap of markers in all photoreceptor subtypes in all time points (~50-70% of the marker genes from our data sets also appear in the published snRNA-seq marker gene list). The overlap is much less when comparing our cone cell and 2°, 3° pigment cell marker genes to the corresponding marker list from the published dataset. About 28-39% of the cone marker genes and about 25-27% of the 2°, 3° pigment cell marker genes from our datasets appear in the previously published snRNA-seq marker gene lists. There is a higher overlap between our 1° pigment cell marker list and the published pigment cell marker list (~58-75%). Thus, their pigment cell cluster may be composed mainly of primary pigment cells.

We also checked the expression patterns of specific marker genes identified from our data sets on the previously published snRNA-seq data. The *Rhodopsin* gene expression patterns in the match well between the two data sets. However, there are substantial differences in cell type specific marker expression. First, *CG2082*, a novel R8 marker gene identified and *in vivo* validated in our work, shows low expression in R8 cells in the published snRNA-seq data set (Figure A). Second, cone cell marker genes identified and *in vivo* validated in our work are not expressed in the cone cell clusters as annotated in the published snRNA-seq data set (Figure B). Third, 2°, 3° pigment cell markers identified and *in vivo* validated in our work are not expressed in the annotated pigment cell cluster in the published snRNA-seq data set (Figure C).

In our Discussion, we have included a statement that our photoreceptor marker genes largely agree with the published snRNA-seq data set but there are some differences between the photoreceptor marker gene lists between the datasets. Moreover, the cone cell and pigment cell marker genes from the published snRNA-seq and our scRNA-seq datasets do not show the same degree of overlap as the photoreceptor markers. Specifically, our validated cone and pigment cell marker genes are not expressed in the cone and pigment cell clusters in the published snRNA-seq dataset.

Gene Expression Plots generated from the published Li et al snRNA-seq data using the web-based tool provided by FlyCellAtlas (Li et al., 2022; https://scope.aertslab.org/#/FlyCellAtlas/FlyCellAtlas%2Fs_fca_biohub_head_10x.loom/gene/). A) R8 cells are shown in red and *CG2082* expression in green. B) Cone cells are in red and cone cell markers, *CG5597* and *CG17211*, in green and blue, respectively. C) Pigment cells are in red, the 2°, 3° pigment cell marker gene, *santa-maria*, is in green, and *wrapper*, the 1° pigment cell marker gene, is in blue.

Reference

Li, H. *et al.* Fly Cell Atlas: A single-nucleus transcriptomic atlas of the adult fruit fly. *Science* **375**, eabk2432, doi:10.1126/science.abk2432 (2022).

Reviewer #2 (Remarks to the Author):

In this manuscript, Yeung et al. present single cell RNA-Seq datasets on adult *Drosophila* eyes that identifies all known cell types. Adult eyes of several ages were subjected to scRNA-Seq and data was processed using the standard Seurat R Package pipeline and Monocle to compare data from different timepoints. The authors identified known cell types and performed differential expression analysis to identify markers. As expected, new markers are found in the datasets and their expression was validated by using the corresponding GAL4 drivers. For example, there are new markers for R7/8, R1-6, cone cells, and two subpopulations of pigment cells. There were no significant changes in gene expression between timepoints, but single-cell experiments revealed some sex-specific signatures. Monocle analysis was not informative as there was not too much difference between animals of different age. The authors include GO enrichment analysis of photoreceptor markers and find GO terms related to vision and phototransduction, while for pigment cells, they find GO terms related to purine and pteridine processing, both findings are expected and do not provide any new information.

The researchers suggest that their dataset may be useful for uncovering molecular mechanisms for eye development. However, as evident by the static transcriptional profiles across multiple time points, the adult eye is terminally developed and therefore not useful for this purpose.

They further suggest that these new markers may serve cell-type specific roles that are important for the maintenance and function of the eye and whose homologs are relevant to human disease.

However, there is simply not enough evidence to make such a claim. There are published scRNA-seq data on the developing *Drosophila* eye, while the adult eye has been examined in recently published Fly Cell Atlas paper, so there is no novelty in this respect.

Overall, it is not clear what the reader can learn from the author's manuscript that has not been already known. In conclusion, the manuscript merely presents a wild type cell atlas of the adult eye with a conventional Seurat analysis and, thus, presents an entry point for further studies. I believe that we are moved beyond the point that this type of work can be published on its own.

Response

We thank the reviewer for the critique of our manuscript. We have compared the marker gene lists from the published snRNA-seq data (Li et al., 2022) with our own marker gene lists. There is good overlap of marker genes between our scRNA-seq and the published snRNA-seq but we also found significant differences between the two data sets. Some of our *in vivo* validated markers (e.g., *CG2082* for R8, and *santa-maria* for 2° and 3° pigment cells) are not found in the same cell types in published snRNA-seq data set. These observations suggest that our data sets differ from the published snRNA-seq data set and we foresee that the additional marker genes discovered and validated in this manuscript will benefit researchers in many ways. We have included this observation in our Discussion in the revised manuscript. We have also transferred our submission to Communications Biology to better fit the scope of our manuscript.

Reviewer #3 (Remarks to the Author):

The compound eye of the fruit fly *Drosophila melanogaster* and its development is a powerful used model system to address a wide range of biological questions. In the current manuscript Yeung and colleagues provide a single-cell transcriptomic dataset on the compound eye. They hereby identify new genes that are expressed in different subtypes.

The approach of the manuscript is valid, and the data included interesting. However, it makes a rather premature impression. I am generally a strong proponent of single-cell sequencing approaches as they provide detailed insight that could not be uncovered otherwise.

While the data set is new, the same data could in principle be obtained from the fly cell atlas published recently (Li et al., 2022, Science).

It is in the meantime standard in the field of single-cell investigations in model organisms to go beyond a catalog of genes and cell types, particularly for a leading journal like Nature Communications.

My comments should not be taken as critique of the approach nor the data. I feel this is an exciting starting point and the authors already have first candidate genes that could potentially be interesting. I can very well imagine that using this as a entry point to e.g. show that newly identified genes are critical for cell type specific functions the manuscript could become an important contribution.

Response

We thank the reviewer for the critique of our manuscript. We have compared cell type specific marker genes identified from our scRNA-seq data set with the published snRNA-seq (Li et al., 2022). We discovered that although there are overlaps between the published marker gene lists and our cell-type specific lists, there are substantial differences between the two data sets. Some of our *in vivo* validated marker genes (e.g., *CG2082* for R8 and *santa-maria* for 2° and 3° pigment cells) are not present in the published snRNA-seq data set. We have added this observation in our Discussion in our revised manuscript. We have also transferred our submission to Communications Biology to better fit the scope of our manuscript.